# Differential stability of task variable representations in retrosplenial cortex

**Luis M. Franco** [1,2] ✉ **& Michael J. Goard** [1,3,4] ✉

Cortical neurons store information across different timescales, from seconds to years. Although information stability is variable across regions, it can vary within a region as well. Association areas are known to multiplex behaviorally relevant variables, but the stability of their representations is not well understood. Here, we longitudinally recorded the activity of neuronal populations in the mouse retrosplenial cortex (RSC) during the performance of a context-choice association task. We found that the activity of neurons exhibits different levels of stability across days. Using linear classifiers, we quantified the stability of three task-relevant variables. We find that RSC representations of context and trial outcome display higher stability than motor choice, both at the single cell and population levels. Together, our findings show an important characteristic of association areas, where diverse streams of information are stored with varying levels of stability, which may balance representational reliability and flexibility according to behavioral demands.

The mammalian brain experiences plasticity throughout its lifespan due to development, learning, ongoing synaptic turnover, and homeostatic maintenance[1-4]. Nevertheless, it must somehow preserve information about known sensory features of the environment, as well as learned associations that are important for survival. How the brain maintains stable representations of the environment in dynamic cortical neural circuits is not well understood. In the visual cortex, for instance, processing of relatively simple features such as orientation tuning seems very stable across many days, and even weeks[5-8]. Similarly, representations of learned movements in the motor cortex[9,10] and somatosensory representations of repeated stimuli in the barrel cortex[11] exhibit considerable stability across multiple days of experimentation (although some neurons can be highly unstable[12]). In the piriform cortex, by contrast, odor representations constantly drift over days unless odors are frequently experienced[13]. Other cortical regions, such as the auditory cortex, experience a monotonic decay in the similarity of their spontaneous activity, despite a small fraction of cells remaining strongly correlated over multiple days[14]. These changes in single-cell response properties can eventually lead to a representational drift in the population code, even in the face of stable

behavioral output[1-4,15]. One possibility is that redundancy in the population code compensates for the turnover of cellular responses, maintaining stable information and animal behavior over long periods of time[3,8,16].

Although sensory and motor cortices can maintain significant levels of stability across long timescales, association areas, and limbic structures appear less stable. In the posterior parietal cortex, a region important for motor planning and spatial decisions, the activity of neurons informative about task features changes considerably across days[17]. In the hippocampus, which is important for the encoding of spatial variables that depend on the integration of multimodal sensory features, CA1 pyramidal cells are highly dynamic, with many unique cells encoding place fields on each particular day[18,19], although granule cells of the dentate gyrus exhibit more stable activity[19,20]. However, recent findings indicate that in the bat hippocampus, CA1 neurons can stably map the trajectory of highly reproducible flights in familiar environments over days and weeks[21], suggesting that in some cases, CA1 hippocampal neurons can exhibit very stable coding. In fact, another study recently showed that two subpopulations can coexist in CA1, one with more stable coding of environmental contexts, and one

[1]Neuroscience Research Institute, University of California, Santa Barbara, Santa Barbara, CA, USA. [2]Institute of Neuroscience, University of Oregon, Eugene, OR, USA. [3]Department of Molecular, Cellular, and Developmental Biology, University of California, Santa Barbara, Santa Barbara, CA, USA. [4]Department of Psychological & Brain Sciences, University of California, Santa Barbara, Santa Barbara, CA, USA. ✉e-mail: luisfran@uoregon.edu; michael.goard@lifesci.ucsb.edu

with a high remapping rate, sensitive to landmark changes[22]. In the retrosplenial cortex (RSC), a region important for the association of egocentric and allocentric spatial representations[23], as well as for the association between environmental context with motor choice[24–26], a considerable fraction of cells maintain stable retinotopic tuning properties to simple visual stimuli[27]. However, it is not known how stable sensory and motor variables are represented in RSC during the performance of a cognitive task.

In the primary visual cortex, recent findings have shown that different stimuli are encoded with differing levels of stability. Representations of stimuli with simple visual statistics such as drifting gratings are highly stable across several weeks of experimentation[5,6]. However, when the same population of neurons is presented with complex stimuli such as naturalistic movies, their representation is much more dynamic, exhibiting progressive changes in response to the same movie over the course of days to weeks[6,28]. This configuration could potentially preserve fundamental tuning properties while allowing more flexible responses to complex naturalistic stimuli. In association areas, such as posterior parietal and retrosplenial cortices, a similar principle might allow highly stable representations of particular behavioral variables during performance of a learned task.

To investigate the stability of neural representations in the RSC during performance of a cognitive task, head-fixed mice were presented two different virtual T-mazes, each with a distinct wall pattern, that determined the correct left or right turn to obtain water rewards[24]. Using longitudinal two-photon microscopy, we measured calcium responses in thousands of neurons in RSC across five consecutive days. We find that neural activity in correct trials is highly stable for both contexts, with less stability in incorrect trials. Support vector machine classifiers trained on the first day of experimentation and tested on the subsequent days revealed varying stability of task-relevant variables in the same population of RSC neurons, both at the single cell and population level. Together, our results indicate that RSC can reliably store sensory information about the environmental context and trial outcome while exhibiting flexible coding of motor choice, which we propose allows adaptability in the face of changing behavioral demands.

## Results
### Differential stability in RSC activity during memory-guided behavior

To investigate the stability of task-relevant representations in the RSC, we developed a behavioral task requiring mice to associate virtual contexts with a left or right joystick turn in order to obtain liquid rewards (Fig. 1a; Supplementary Movie 1; for a full description of this task, see "Methods" section and ref. 24). We then used two-photon calcium imaging to record neural activity in transgenic mice with pan-excitatory expression of GCaMP6s. We imaged a total of 4701 neurons in layer 2/3 of the dysgranular RSC across 5 consecutive days ($n = 18$ experiments in 6 mice). The same field of view was identified in each session, and video stacks were registered using day 1 as reference (see "Methods" section). To ensure that we had enough data for all subsequent analyses, we only considered sessions in which mice had ≥2 trials of each type (correct-left, incorrect-left, incorrect-right, correct-right), and also maintained a behavioral performance above chance level (Fig. 1b–e; more details in Supplementary Fig. 1). On average, mice made (mean ± s.e.m.): 19.1 ± 1.0 correct left, 7.7 ± 0.7 incorrect right, 6.3 ± 0.5 incorrect left, and 23.0 ± 1.4 correct right decisions (Fig. 1c), which translated to an overall behavioral performance = 74.1% ± 1.0% (mean ± s.e.m.) across all experiments (Fig. 1b and Supplementary Fig. 1). Their performance, response bias and response latency stabilized during training, and was consistent across imaging sessions (Supplementary Fig. 2). In addition, we only considered neurons with significant morphological stability[29], as measured by the structural similarity index (SSI) of their corresponding ROI regions across days

(Fig. 1f, g, Supplementary Fig. 3), and significant reliable responses across days (see "Methods" section). Together, this yielded different overlapping groups of neurons (day 2 = 1868 neurons; day 3 = 1908 neurons; day 4 = 1609 neurons; day 5 = 2020 neurons) whose responses we compared to those in day 1 (3087 neurons with at least one pair in days 2 to 5). Fig. 1h shows two example neurons with preserved activity dynamics across days.

At the population level, RSC neurons appear to maintain a similar structure across days. To visualize this, we plotted the activity of the neural population sorted by cross-validated latency, using odd trials in day 1 to sort neural responses by latency, and then plotting the responses in even trials across days (Fig. 2a). We observe that responses in correct trials are more consistent across days, with a similar ridge in population activity (yellow diagonal, Fig. 2a). To test this, we calculated the Pearson's correlation coefficient of population vectors in correct and incorrect trials (Fig. 2b). We find that activity in correct trials is indeed more stable than in incorrect trials, as revealed by higher correlations between cross-validated population vectors (Fig. 2b), which are significantly higher along the diagonal describing the trial duration (Fig. 2c; $n = 1609$ neurons resampled over 100 iterations; mean correlation in correct trials >95th CI of the correlation in incorrect trials). To ensure this was not due to unequal trial numbers, we sampled equally across trial types and confirmed that population vectors in correct trials exhibit higher stability (Supplementary Fig. 4). In addition, we noticed that population activity in correct trials experienced a steeper decay in stability after one day, and a slower decay on the following days, but maintaining substantial levels of structure (Fig. 2a) and stability (Fig. 2b, c) across the 5 days of experimentation. This suggests that although activity in individual neurons undergoes different levels of drift, the population preserves considerable task-related information across days, particularly in correct trials. Moreover, when we analyzed the responses of RSC neurons in each day independently, we found similar levels of activity (Supplementary Fig. 5a), a similar fraction of neurons mapping the trial duration (Supplementary Fig. S5b), and similar preference indices for a particular context (Supplementary Fig. 5c). We also observed similar levels of activity in correct and incorrect trials (Supplementary Fig. 5d, e). This indicates that while some neurons lose their task-driven responses across days (Fig. 2a), other neurons become tuned to the task on each day (Supplementary Fig. 5a), contributing to the homeostasis of task-related representations across the population. Moreover, these results indicate that population dynamics are preserved in the RSC circuit across days, despite some variability in single-cell activity.

Although population responses were fairly stable in correct trials, this does not necessarily mean that the responses of individual neurons to specific task epochs are stable across days. To further investigate the drift in the activity of individual cells, we compared responses exhibiting a significant difference between the two visual environments (context 1 or context 2), the two motor choices (left or right), or the two trial outcomes (rewarded or not rewarded), using balanced numbers of correct and incorrect trials to distinguish correlated task variables (see "Methods" section). Fig. 3a shows example cells with significant activity difference between contexts 1 and 2 (cell 1), between left and right choices (cell 2), and between rewarded and not rewarded trials (cell 3, before trial end; cell 4, after trial end). Supplementary Figure 6 contains additional example cells. Note that responses tuned to context and trial outcome appear more stable than responses tuned to motor choice in these example cells (Fig. 3a and Supplementary Fig. 6). To quantify this, we calculated the Pearson's correlation coefficient between responses grouped by their preferred context, motor choice or trial outcome, comparing odd trials in day 1 against even trials in days 1 to 5 (Fig. 3b, c). We find that responses grouped by context tend to be slightly more stable than responses grouped by motor choice, and that responses grouped by trial

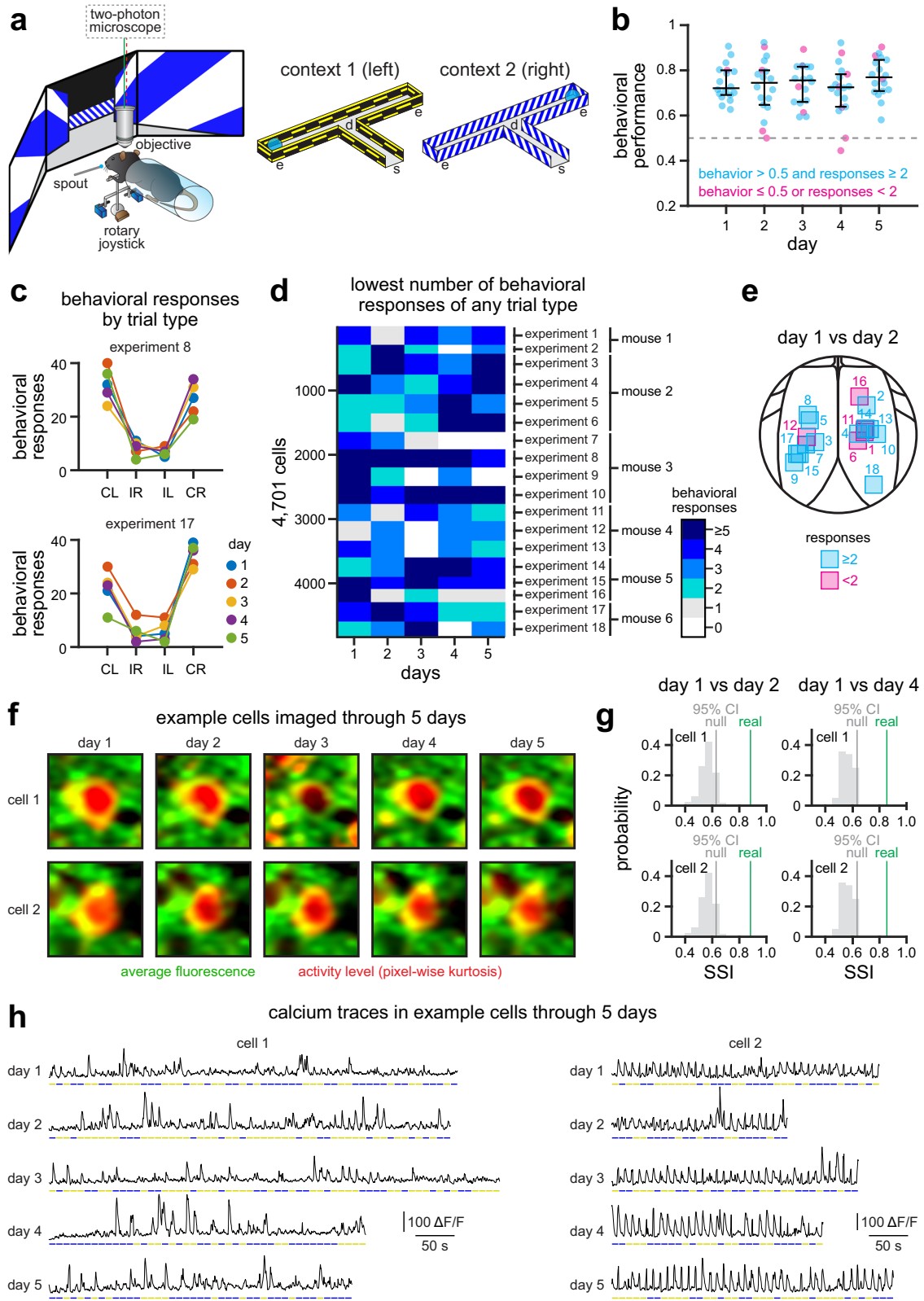

outcome (either before or after trial end) show the highest stability (Fig. 3c; $CC_{day1}$, $p_{C-M} = 0.2$, $p_{C-O1} \leq 10^{-5}$, $p_{C-O2} \leq 10^{-5}$, $p_{M-O1} \leq 10^{-5}$, $p_{M-O2} \leq 10^{-5}$, $p_{O1-O2} = 0.95$; $\tau$, $p_{C-M} = 0.98$, $p_{C-O1} \leq 10^{-5}$, $p_{C-O2} = 1.1 \times 10^{-5}$, $p_{M-O1} \leq 10^{-5}$, $p_{M-O2} = 2.6 \times 10^{-3}$, $p_{O1-O2} = 0.08$; data were fit for each individual cell, their model parameters were rank transformed and then compared using linear mixed-effects (LME) models, with a fixed effect for task variable, and random effects for mouse and experimental

session; see "Methods" section for details). However, post-decision and post-trial outcome responses are difficult to disentangle through correlation analysis, since many neurons respond to both, and are investigated further in the next section. In addition, we find that responses in correct trials are more correlated with each other than responses in incorrect trials (Fig. 3d; $CC_{day1}$, $p_{CL-IR} \leq 10^{-5}$, $p_{CL-IL} \leq 10^{-5}$, $p_{CL-CR} = 0.02$, $p_{IR-IL} = 5.1 \times 10^{-3}$, $p_{IR-CR} \leq 10^{-5}$, $p_{IL-CR} \leq 10^{-5}$; $\tau$, $p_{CL-IR} \leq 10^{-5}$, $p_{CL-IL} \leq 10^{-5}$,

**Fig. 1 | Context-choice association task for studying stability of cortical representations in mice. a** Experimental setup. Mice use a rotary joystick to choose left or right trajectories depending on the visual context. Neural activity is recorded using a two-photon microscope[23]. s = trial start; d = decision point; e = trial end. **b** Behavioral performance across 5 consecutive days ($n = 18$ experiments in 6 mice). Performance remained above chance in all experiments (median): day 1 = 72.1%, day 2 = 74.5%, day 3 = 75.5%, day 4 = 72.5%, day 5 = 76.9%. Dots, individual sessions; black overlay, median ± 75% CIs. **c** Example experiments showing the number of behavioral responses across 5 days of experimentation (more examples in Supplementary Fig. 1a). CL correct left, IR incorrect right, IL incorrect left, CR correct right. **d** Heatmap showing the lowest number of behavioral responses of any type in each experiment. For instance, on day 4 experiment 8 (Fig. 1c) there are 29 CL, 9 IR, 7 IL, and 34 CR responses. Thus, the lowest number of behavioral

responses is 7. Only experiments with at least 2 responses from each trial type are considered for further analyses. **e** Map showing the location of included (cyan) and excluded (magenta) experiments for all comparisons between days 1 and 2. See Supplementary Fig. 1b, c for maps of all comparisons between days. **f** Example cells imaged across 5 days of experimentation (30 × 30 μm field). Note the stability of the imaging field (green), regardless of variability in activity levels (red). **g** We estimated the stability of ROIs by calculating the SSI between 30 μm square regions surrounding each ROI across days, and compared that to a null distribution of SSIs (see "Methods" section; Supplementary Fig. 3). Only ROIs with a SSI higher than the 95th CI of their null SSI were considered for further analyses. **h** Activity traces for two example cells. Note conserved dynamics across days (cell 1 principally responds in context 1; cell 2 responds in both contexts). Source data for B, C, D, and G are provided as a Source Data file.

$p_{\text{CL-CR}} = 0.31$, $p_{\text{IR-IL}} = 0.19$, $p_{\text{IR-CR}} \leq 10^{-5}$, $p_{\text{IL-CR}} \leq 10^{-5}$; data were fit for each individual cell, the model parameters were ranked and then compared using LME models; fixed effect for trial type, random effects for mouse and experimental session), confirming our observations in the population data (Fig. 2). This suggests that most task-related information is conserved in correct trials across days. However, correlations between responses in incorrect trials consistently stayed above zero, suggesting a contribution in the stability of task-relevant representations (Fig. 3d). In terms of their response dynamics, both context and motor-related responses exhibit similar profiles in correct and incorrect trials, with more prominent responses typically between trial start and trial end (cells 1 and 2 in Fig. 3a; cells 5 and 6 in Supplementary Fig. 6). Interestingly, responses grouped by trial outcome can either peak after the decision point or after trial end, with some cells exhibiting mixed dynamics (cells 3 and 4 in Fig. 3a; cells 7 and 8 in Supplementary Fig. 6). Thus, for all subsequent analyses we consider these two outcome-related responses separately, as they likely underlie different cognitive processes. Overall, our analyses of calcium responses indicate that activity in the RSC circuit is partially stable, with conserved population structure across days, particularly in correct trials (Fig. 2), which likely depends on the specific tuning properties of each cell (Fig. 3). However, although the response correlation is useful to draw general comparisons between activity traces in individual neurons or between population vectors, a more detailed examination that considers the structure of the trial is necessary in order to determine whether representations of specific task variables exhibit distinct levels of stability.

### Encoding of task variables exhibits distinct levels of stability in individual RSC neurons

The analysis of responses across days indicates that responses to particular task variables are unstable across days, and may show differential stability between task variables. However, response correlation might be biased for transient responses (e.g., outcome responses) and not account for changes in coding strength across the entire trial. To assess the stability in the encoding of task variables throughout the time course of the trial, we evaluated the performance of linear decoders across each individual time point[24,30]. To this end, we trained support vector machine classifiers for each specific task variable (context, choice, outcome) using activity in day 1 as reference, and tested the performance of these classifiers on the activity of the same cells in the following days (see "Methods" section for details). To analyze day 1 activity without overfitting, we used half of the trials for fitting the classifier, and the other half for testing. We find that some neurons specifically and significantly encode task-related information in day 1 and, in some cases, can maintain their encoding preference across days with similar temporal dynamics as in day 1 (Fig. 4a; more examples in Supplementary Fig. 7; for details about defining significant encoding see Supplementary Fig. 8). However, it appears different task variables are encoded with

different levels of stability. From the examples in Fig. 4a and Supplementary Fig. 7, we observe that post-trial outcome (cells 4, 8, and 12) and context (cells 1, 5, and 9) neurons exhibited the highest coding stability. Post-decision outcome (cells 3, 7, and 11) neurons also show some level of coding stability, while motor choice (cells 2, 6 and 10) neurons display a rapid decay. Indeed, when we look at the population of neurons displaying significant encoding of each task variable in day 1, we observe a similar trend, with outcome and context neurons sustaining higher decoding performance than motor neurons (Fig. 4b). This is also confirmed by the higher average decoding performance of neurons that significantly encode outcome and context across days (Fig. 4c; perf$_{\text{day1}}$, $p_{\text{C-M}} = 0.68$, $p_{\text{C-O1}} = 0.76$, $p_{\text{C-O2}} \leq 10^{-5}$, $p_{\text{M-O1}} = 0.67$, $p_{\text{M-O2}} \leq 10^{-5}$, $p_{\text{O1-O2}} \leq 10^{-5}$; $\tau$, $p_{\text{C-M}} = 7.8 \times 10^{-3}$, $p_{\text{C-O1}} = 0.57$, $p_{\text{C-O2}} \leq 10^{-5}$, $p_{\text{M-O1}} = 5.1 \times 10^{-4}$, $p_{\text{M-O2}} \leq 10^{-5}$, $p_{\text{O1-O2}} \leq 10^{-5}$; data were fit for each individual cell, the model parameters were ranked and then compared by LME models; fixed effect for task variable, random effects for mouse and experimental session). Our results suggest that the RSC constantly experiences a representational drift in the storage and encoding of task-related information, with variables exhibiting different drift rates.

One implication of the observed progressive decay in the encoding of information across days is that task information will eventually be lost by the circuit, given enough time. Yet, we observe that mice maintain behavioral performance for several weeks[24]. For this reason, we trained and tested linear classifiers in each day of experimentation independently. We find consistent subpopulations of neurons encoding context, motor choice, post-decision outcome, and post-trial outcome in each particular day (Supplementary Fig. 9). This suggests that although some neurons lose their ability to encode features of the task, other neurons show increased task variable encoding, maintaining homeostasis in the RSC circuit.

Our analysis of encoding of task information in individual neurons revealed another important finding. We observe that although most neurons specifically encode information about a single task variable, some neurons can multiplex different sets of variables (Fig. 4d and Supplementary Fig. 10). For this reason, we investigated how different types of information within individual neurons are preserved over time. To evaluate this, we compared the performance of support vector machine classifiers in neurons with significant encoding of two task variables on day 1 (Fig. 5a, b and Supplementary Fig. 11a, b). By examining the population of neurons, we find a larger fraction of cells that maintain their encoding of context and outcome information than their encoding of motor choice (Fig. 5a and Supplementary Fig. 11a). This trend can also be observed in individual examples (Fig. 5b and Supplementary Fig. 11b). We confirmed these findings by calculating the decoding performance across days (Fig. 5c; perf$_{\text{day1}}$, $p_{\text{C-M}} = 0.23$, $p_{\text{C-O1}} = 0.01$, $p_{\text{C-O2}} \leq 10^{-5}$, $p_{\text{M-O1}} \leq 10^{-5}$, $p_{\text{M-O2}} \leq 10^{-5}$, $p_{\text{O1-O2}} \leq 10^{-5}$; $\tau$, $p_{\text{C-M}} = 0.21$, $p_{\text{C-O1}} = 0.10$, $p_{\text{C-O2}} \leq 10^{-5}$, $p_{\text{M-O1}} \leq 10^{-5}$, $p_{\text{M-O2}} \leq 10^{-5}$, $p_{\text{O1-O2}} \leq 10^{-5}$; data were fit for each individual cell, the model parameters were ranked and then compared by LME models; fixed effect

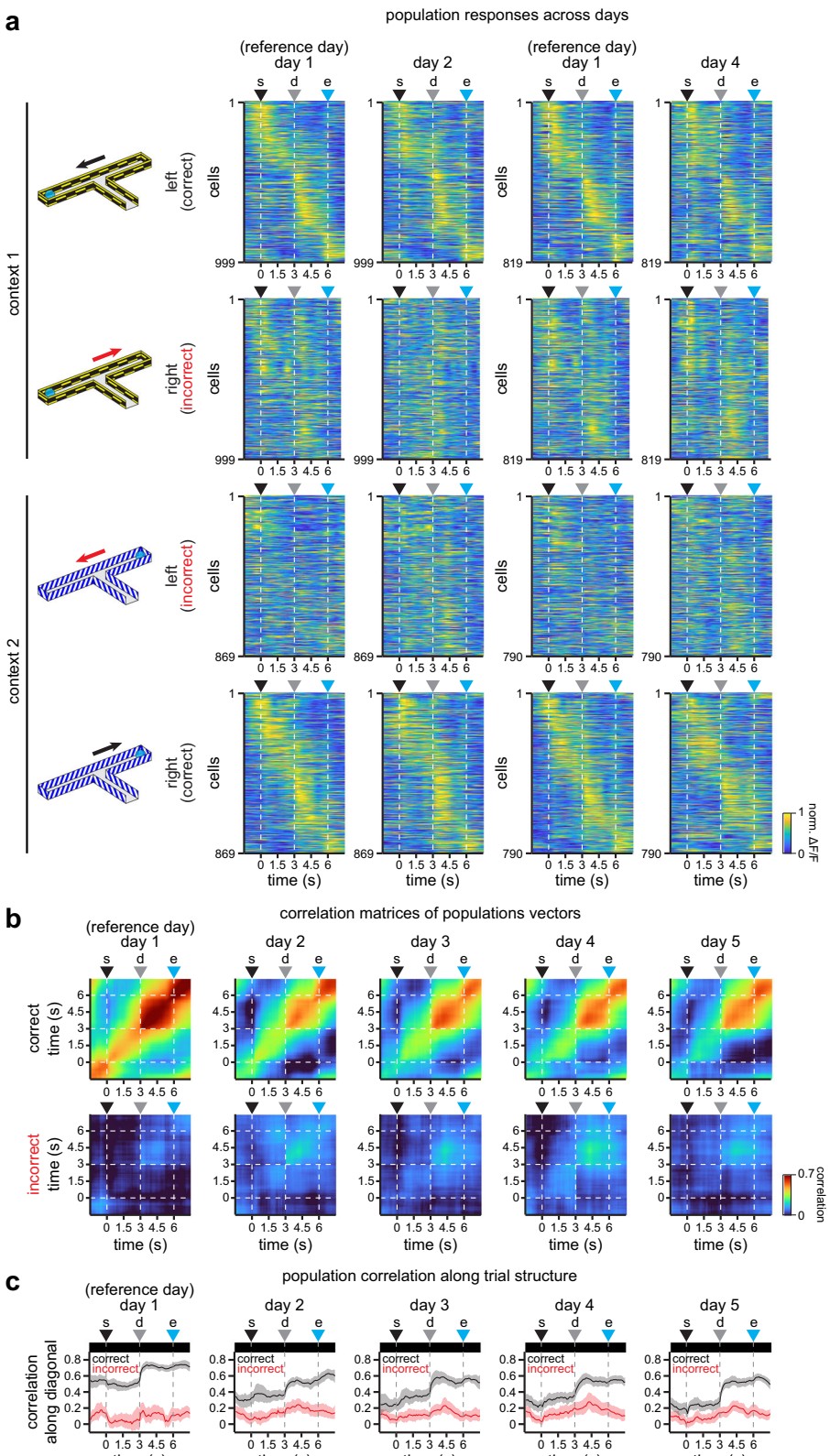

for task variable, random effects for mouse and experimental session). Moreover, of the two different types of outcome encoding, post-trial outcome showed higher stability (Supplementary Fig. 12), which might be related to the presence of a reward and/or increased licking in correct trials. To further characterize neurons multiplexing different task variables, we compared the density of the area under the curve of the classifier performances (Supplementary Fig. 13a) for

the subpopulations shown in Fig. 5a and Supplementary Fig. 11a. We observe higher stability in the encoding of context in context-motor cells, particularly in the first half of the trial, before the decision point (top row, Supplementary Fig. 13a, b). Similarly, we observe higher stability in the encoding of trial outcome in motor-outcome cells, specifically after the decision point during the second half of the trial (two bottom rows, Supplementary Fig. 13a, b). In context-outcome

**Fig. 2 | Population activity in RSC exhibits different levels of stability in correct and incorrect trials. a** Average normalized activity in the recorded population of neurons separated by their preferred context. For clarity, only comparisons between days 1 and 2, and days 1 and 4 are shown. Activity is cross-validated, sorted by latency using odd trials in day 1, and plotted using even trials for days 2 or 4. **b** Pearson's correlation coefficient matrices showing the similarity of population activity along trial duration (averaged across 100 iterations of cross-validated population vectors constructed from the preferred responses in each context, as shown in Fig. 2a). Briefly, population vectors in one half of resampled trials in day 1 are compared to population vectors in the other half of resampled non-overlapping

trials in day 1. For the other days, all resampled trials in day 1 are compared to all resampled trials in days 2 to 5. In addition, to compare vectors of the same length across days, the population is resampled to 1609 cells within each iteration (the lowest number of cells that meet criteria in all 5 days). **c** Population vector correlation coefficients along the diagonals in Fig. 2b ($n = 100$ iterations). Note the higher stability and similarity of population activity in correct trials. Top insets indicate time bins (in black) where the mean correlation in correct trials >95th CI of the correlation in incorrect trials throughout the trial. Solid line, mean; shaded area, 95% CIs. For all panels: s trial start, d decision point, e trial end.

cells, however, stability seems bimodal, with context showing higher stability before the decision point, and trial outcome higher stability by the end of the trial (second and third rows, Supplementary Fig. 13a, b). These results highlight an important feature of RSC neurons that multiplex task information, which favors the maintenance of environmental context and trial outcome over the encoding of motor choice.

## Differential stability in population coding of task variables in the RSC

Our findings on the stability of population activity (Fig. 2b, c) and, in particular, on the differential stability of information encoding in individual neurons (Fig. 4b, c) suggest there may be similar dynamics in the population-level encoding of task variables in the RSC. To test this hypothesis, we trained support vector machine classifiers on population vectors in day 1, and tested those models on population vectors in the following days for each individual 5-day experiment. Inspection of an example experiment (Fig. 6a) as well as the entire group of experiments (Fig. 6b) further support our previous findings, in which encoding of context and trial outcome exhibit the highest stability while encoding of motor choice shows lower stability. This is also confirmed by fitting an exponential decay function to the performance of classifiers in the group of experiments shown in Fig. 6b. Briefly, we calculate the area between the classifier performance curve and the chance level (0.5) at relevant windows during task performance for each day: 1.5–3 s for environmental context (immediately prior to the decision point), 3–4.5 s for motor choice and post-decision outcome (immediately after the decision point), and 6–7.5 s for post-trial outcome (during the reward period). We then normalize the area in day 1 (when the classifier is trained), and fit an exponential decay function ($NP = e^{-1/\tau}$) to estimate the time at which decoding performance decays to 1/e (-37%) of its original value in day 1. The normalized performance (NP) allows us to measure the encoding stability of task variables, where a value = 1 signifies same decoding performance as in day 1 and a value = 0 indicates that decoding performance has decreased to chance level. Consistent with our observations in individual neurons, we find that context ($\tau = 4.3$ days) and post-trial outcome ($\tau = 5.1$ days) exhibit the highest stability, whereas post-decision outcome ($\tau = 3.1$ days) and motor choice ($\tau = 1.6$ days) display lower stability (Fig. 6c). Moreover, fitting exponential decays to each individual experiment revealed similar results (Fig. 6d, Supplementary Fig. 14a, b; $n = 8$ [context], 11 [motor], 11 [post-decision outcome], 11 [post-trial outcome]). Pairwise variable comparisons using LME models found significant differences between context and motor ($p = 0.02$), motor and post-decision outcome ($p = 0.01$), motor and post-trial outcome ($p \leq 10^{-5}$), and between post-decision and post-trial outcome ($p = 0.01$), further supporting differential stability of task-related representations in the RSC, and its persistence within single experiments and across the population of experiments. Similarly, data grouped by mouse showed qualitatively similar results (Supplementary Fig. 14c, d). Importantly, this representational drift might indicate that individual experiments in some mice no longer encode task variables after a few days. However, models trained and tested on each day independently

showed that there is sufficient coding power in the population throughout the duration of the experiments (Supplementary Fig. 15).

In order to reduce sampling variability in individual experiments and to evaluate whether the encoding of task variables exhibit similar dynamics across the RSC circuit, we used pseudo-populations of neurons pooled across experimental sessions to analyze the data across all experiments and all mice. In accordance with our previous findings[24], classifiers trained on population vectors show high decoding performance of environmental context between trial start and trial end, which corresponds to the presence of visual stimuli (upper left panel, Fig. 7a). Decoding of motor choice is highest around the decision point, when mice rotate the joystick to select a trajectory through the maze (top-middle left panel, Fig. 7a). Decoding of post-decision outcome exhibits a prominent peak around the decision point (bottom-middle left panel, Fig. 7a), whereas decoding of post-trial outcome displays ceiling level performance during the reward period, after trial end (bottom left panel, Fig. 7a). In terms of their stability, decoding of environmental context shows a clear and smooth decay across the 5 days of experimentation, with performance above chance level (0.5, dashed horizontal line) even on day 5 (top row, Fig. 7a, b). By contrast, population decoding of motor choice undergoes a rapid decay from day 1 to day 2, and declines to near chance levels by day 3 (top-middle row, Fig. 7a, b). In addition, post-trial outcome exhibits very stable population decoding at the end of the trial (6–7.5 s) across the 5 days of experimentation (bottom row, Fig. 7a, b), whereas post-decision outcome (3–4.5 s) displays a more rapid decay (bottom-middle row, Fig. 7a, b). To quantify this, we fit exponential decay functions to the performance of classifiers trained on population data (Fig. 7c). Consistent with our observations in single cells (Figs. 4 and 5) and individual experiments (Fig. 6), we find that context ($\tau = 7.3$ days; 95% CIs = 5.9–9.7 days) and post-trial outcome ($\tau = 35.7$ days; 95% CIs = 25.9–50.0 days) are considerably more stable than motor choice ($\tau = 1.9$ days; 95% CIs = 1.4–2.5 days) and post-decision outcome ($\tau = 2.2$ days; 95% CIs = 1.9–2.5 days), as shown by the non-overlap between their 95% CIs (Fig. 7c).

To ensure our findings can be generalized regardless of the chosen reference day, we also trained support vector machine classifiers on population data from days 2 to 5 and tested them on the reminder of days (Supplementary Fig. 16). We find a similar trend, with representation of context and post-trial outcome being considerably more stable than representations of motor choice and post-decision outcome, even when evaluating models in future or past days (Supplementary Fig. 16c). In addition, since we maximized the usage of available data throughout this study based on behavioral parameters (Fig. 1b–e and Supplementary Fig. 1), morphological stability of ROIs (Fig. 1f, g and Supplementary Fig. 3), and reliability of task-related neuronal activity, we then analyzed only neurons that pass all these tests consecutively across all 5 days of experimentation. We confirmed that this smaller subpopulation of 601 neurons also exhibits a similar differential stability profile in task-related representations of the RSC (Supplementary Fig. 17). Training and testing support vector machine classifiers on the inferred spikes obtained by deconvolution of calcium signals (Supplementary Fig. 18a, b), also reveals similar dynamics in

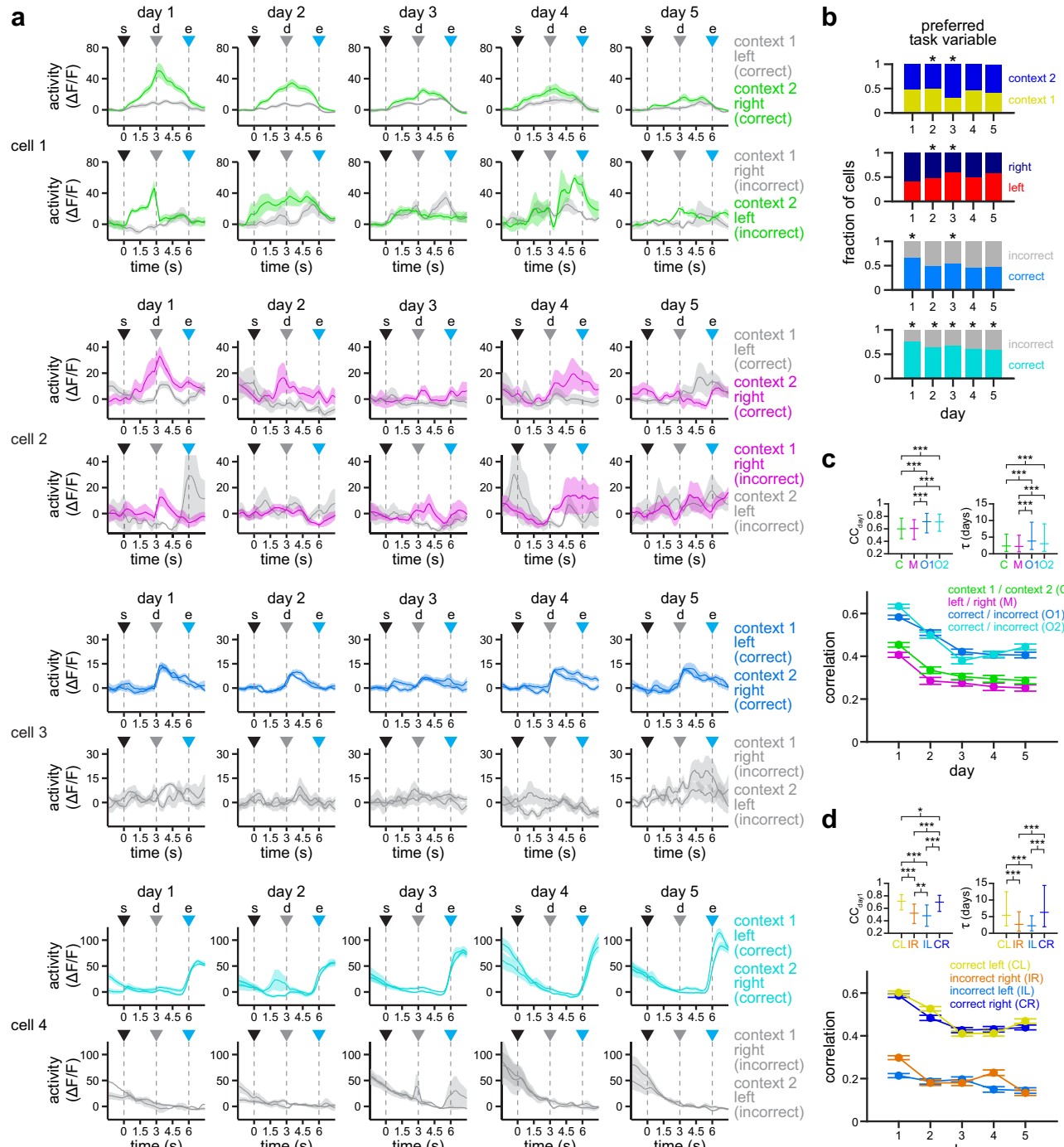

**Fig. 3 | Activity in individual RSC neurons tuned to visual context, motor choice or trial outcome exhibit different levels of stability. a** Example neurons exhibiting higher responses for context 2 (cell 1), for right turns (cell 2), for correct post-decision outcome (cell 3), and for correct post-trial outcomes (cell 4), respectively. Traces are color-coded to highlight the difference between the preferred and non-preferred responses in both correct (top row) and incorrect (bottom row) trials. Solid line, mean; shaded area, s.e.m. s trial start, d decision point, e trial end. **b** Fraction of cells with a preference either context, motor choice or outcome (evaluated before or after trial end). Preference is defined as higher activity in the indicated subset of trials. Briefly, trials are equally sampled within each subset, then compared using a two-sided Wilcoxon signed-rank test. If activity is significantly different over 5 consecutive time bins (0.5 s), then a cell is considered to have a significant preference. Asterisks denote differential fraction of cells, assessed by

non-overlap of the bootstrapped 95% CI ranges. **c**, **d** Average Pearson's correlation coefficients (CC) between odd trials in day 1 and even trials in days 1 to 5 (mean ± s.e.m.). **c** shows data grouped by their preferred task variable ($n = 1,181, 669, 791, 574, 795$ cells in days 1–5 [context]; $n = 1158, 628, 763, 529, 785$ cells in days 1–5 [motor]; $n = 1,527, 841, 1045, 814, 1040$ cells in days 1–5 [post-decision outcome]; $n = 1161, 601, 789, 666, 805$ cells in days 1–5 [post-trial outcome]), whereas **d** shows data grouped by trial type ($n = 1570, 999, 979, 819, 1055$ cells in days 1–5 [correct left and incorrect right]; $n = 1517, 869, 929, 790, 965$ cells in days 1–5 [incorrect left and correct right]). Top insets show significantly different comparisons for fit parameters (*$p < 0.05$, **$p < 0.01$, ***$p < 0.001$; two-sided F-test calculated by LME models; see "Methods" section for details). Source data for B, C, and D are provided as a Source Data file.

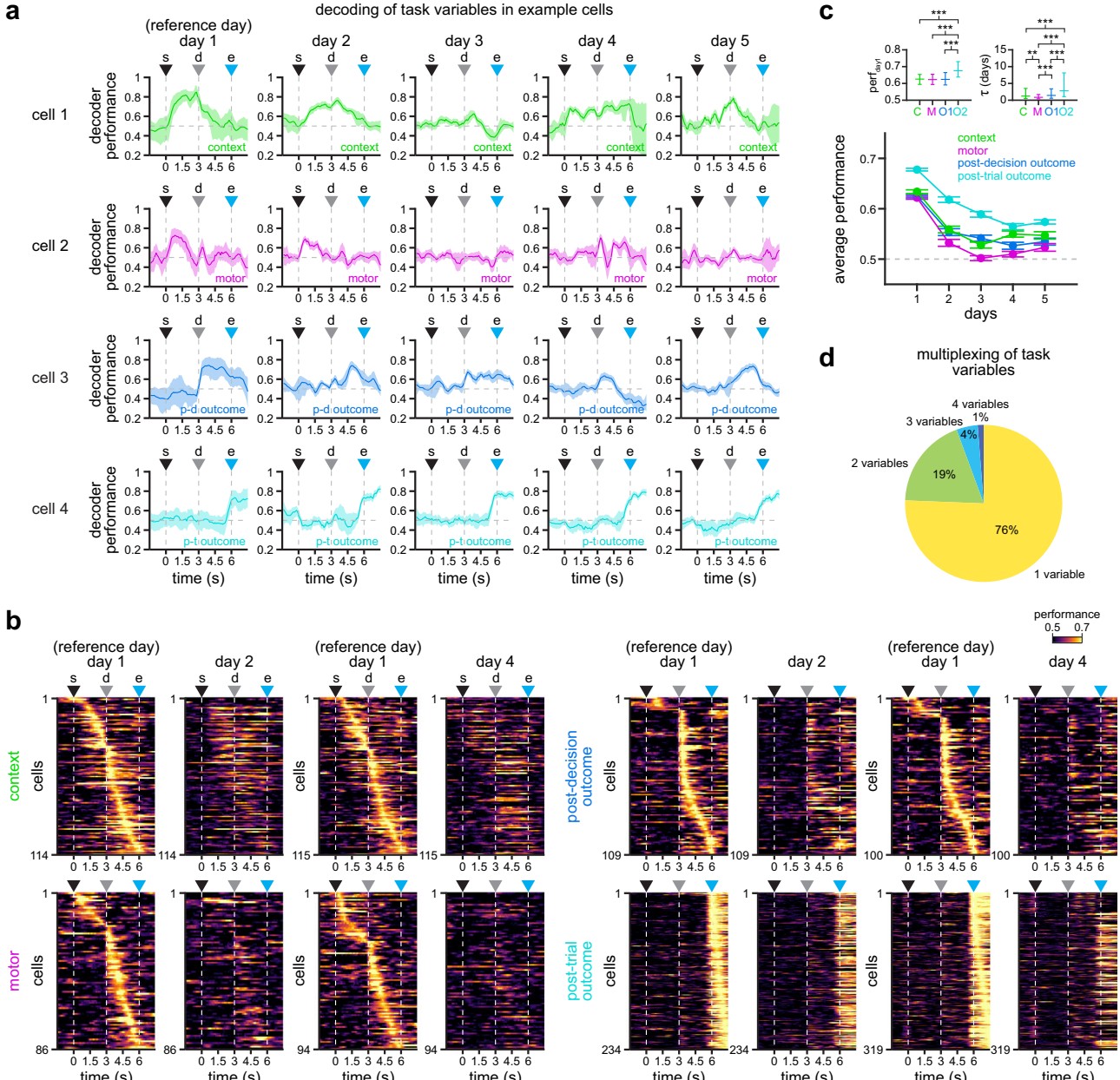

**Fig. 4 | RSC neurons encode task variables with differential stability. a** Example cells displaying significant decoding of context (cell 1, top), motor choice (cell 2, top-middle), post-decision outcome (cell 3, bottom-middle), and post-trial outcome (cell 4, bottom) in day 1. Solid line, mean; shaded area, 95% CIs. Decoding performance obtained using support vector machine classifiers. **b** Average decoding of the indicated task variables in single cells. All cells exhibit significant decoding in day 1. Only comparisons to days 2 and 4 are shown for clarity. Note that decoding of outcome and context is more stable than decoding of motor choice. **c** Average decoding performance of cells that show significant decoding in day 1

(mean ± s.e.m.; n = 197, 114, 129, 115, 121 cells in days 1–5 [context]; n = 173, 86, 120, 94, 126 cells in days 1–5 [motor]; n = 164, 109, 115, 100, 107 cells in days 1–5 [post-decision outcome]; n = 488, 234, 324, 319, 353 cells in days 1–5 [post-trial outcome]). Top inset shows significantly different comparisons for fit parameters (*p < 0.05, **p < 0.01, ***p < 0.001; two-sided F-test calculated by LME models; see "Methods" section for details). **d** Percentage of cells with significant decoding of 1, 2, 3, or 4 task variables in day 1. For all panels: s trial start, d decision point, e trial end. Source data for C are provided as a Source Data file.

RSC representations, with context and post-trial outcome displaying higher stability across days (Supplementary Fig. 18c, d). Moreover, we confirmed that our stability analyses are not biased by the normalization of decoding performance, which allowed us to estimate decay constants for the decline in decoding performance across days. The average performance of classifiers trained on population data also shows higher stability for the context and outcome variables, and lower stability for motor choice at behaviorally relevant windows (Supplementary Fig. 19a). In addition, since the size of the

subpopulations of neurons exhibiting significant decoding in day 1 can vary (Fig. 4b), we trained classifiers with equally sampled numbers of neurons. We again find higher stability in the encoding of context and post-trial outcome (Supplementary Fig. 19b). To ensure that greater stability is not solely a feature of higher day 1 performance, we balanced day 1 performance by determining the number of randomly sampled neurons necessary for 90% decoding of each task variables on day 1 (Supplementary Fig. 19c). Using these subpopulations, we once again found higher stability for context and post-trial outcome, and

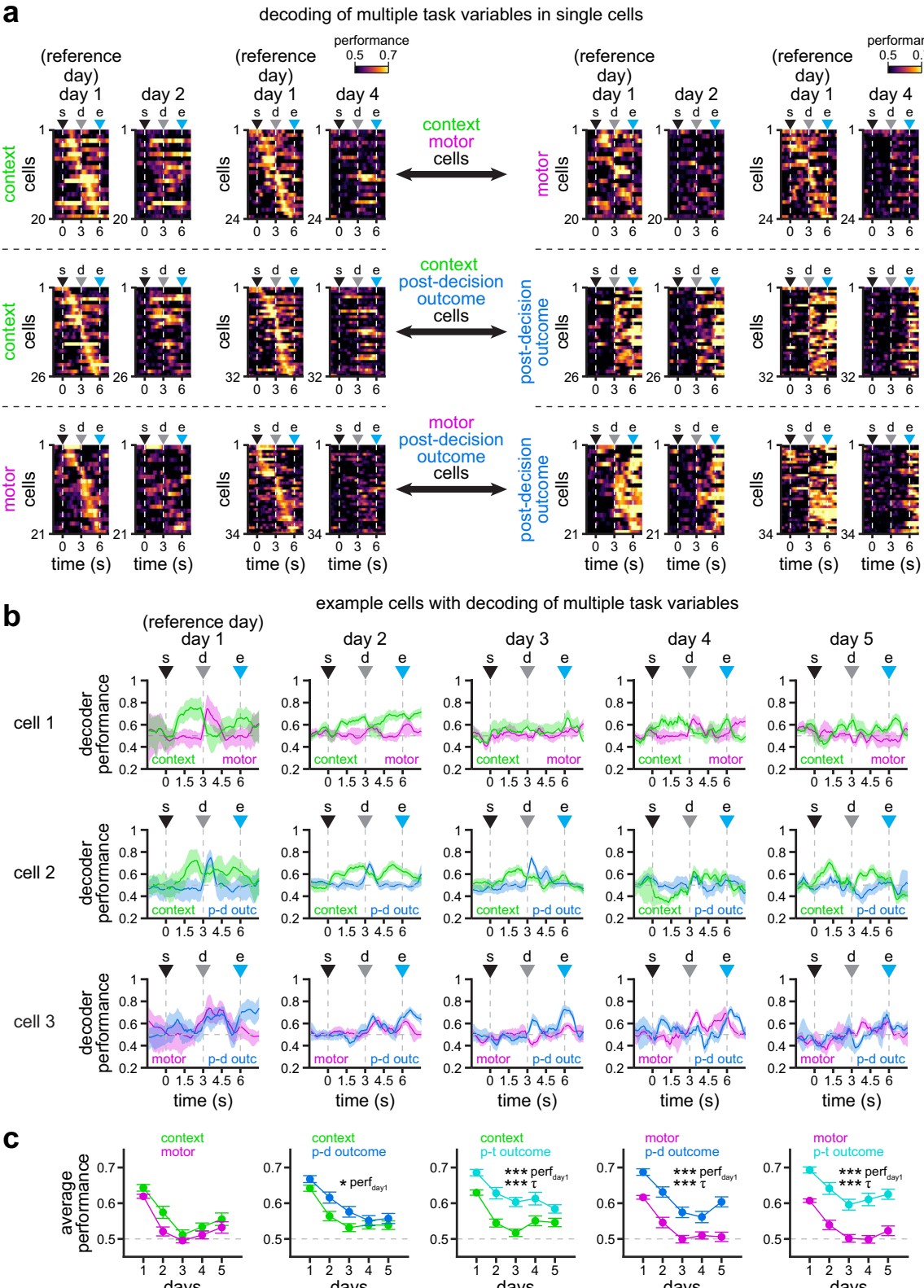

**a** decoding of multiple task variables in single cells

**b** example cells with decoding of multiple task variables

**c**

lower stability for motor choice and post-decision outcome (Supplementary Fig. 19d).

Together, our results show a gradient in the stability of information encoding and storage in the RSC, in which environmental context is more stable than motor choice. This, in turn, suggests a mechanism where the RSC reliably stores sensory information from the environment while, at the same time, allowing flexibility in the mapping of

motor choice, which likely depends on current sensory demands and reward history (Fig. 8a). Moreover, it is likely that association areas, which multiplex different streams of information, encode sensory and motor signals with lower levels of stability than the dedicated sensory and motor cortices they connect to (Fig. 8b). This ability of association areas to process information in a more flexible manner than dedicated sensory and motor cortices might help to associate incoming sensory

**Fig. 5 | Multiplexing of task information with varying stability within single RSC neurons. a** Average performance of support vector machine classifiers on the activity of single cells that significantly encode two task variables: context and motor choice (top row), context and post-decision outcome (middle row), and motor choice and post-decision outcome (bottom row) in day 1. Additional cells with multiplexing decoding of different pairs of variables are shown in Supplementary Figs. 11 and 12. Note the better decoding performance of models trained in day 1 and tested in days 2 and 4 for context and outcome variables. **b** Example cells with significant encoding of context and motor choice (cell 1, top), context and post-decision outcome (cell 2, middle), and motor choice and post-decision outcome (cell 3, bottom) in day 1. Solid line, mean; shaded area, 95% CIs. **c** Average

decoding performance of the two task variables indicated in each plot within individual cells (mean ± s.e.m.; $n = 43, 23, 40, 35, 40$ cells in days 1–5 [context-motor]; $n = 48, 31, 42, 44, 44$ cells in days 1–5 [context-post-decision outcome]; $n = 55, 33, 51, 51, 53$ cells in days 1–5 [context-post-trial outcome]; $n = 52, 30, 49, 47, 49$ cells in days 1–5 [motor-post-decision outcome]; $n = 55, 31, 51, 49, 51$ cells in days 1–5 [motor-post-trial outcome]). Asterisks denote significantly different decoding performance, as measured by different fit (*$p < 0.05$, **$p < 0.01$, ***$p < 0.001$; two-sided F-test calculated by LME models; see "Methods" section for details). Note the higher decoding performance of the classifiers decoding context and outcome when compared to motor choice. For all panels: s trial start, d decision point, e trial end. Source data for C are provided as a Source Data file.

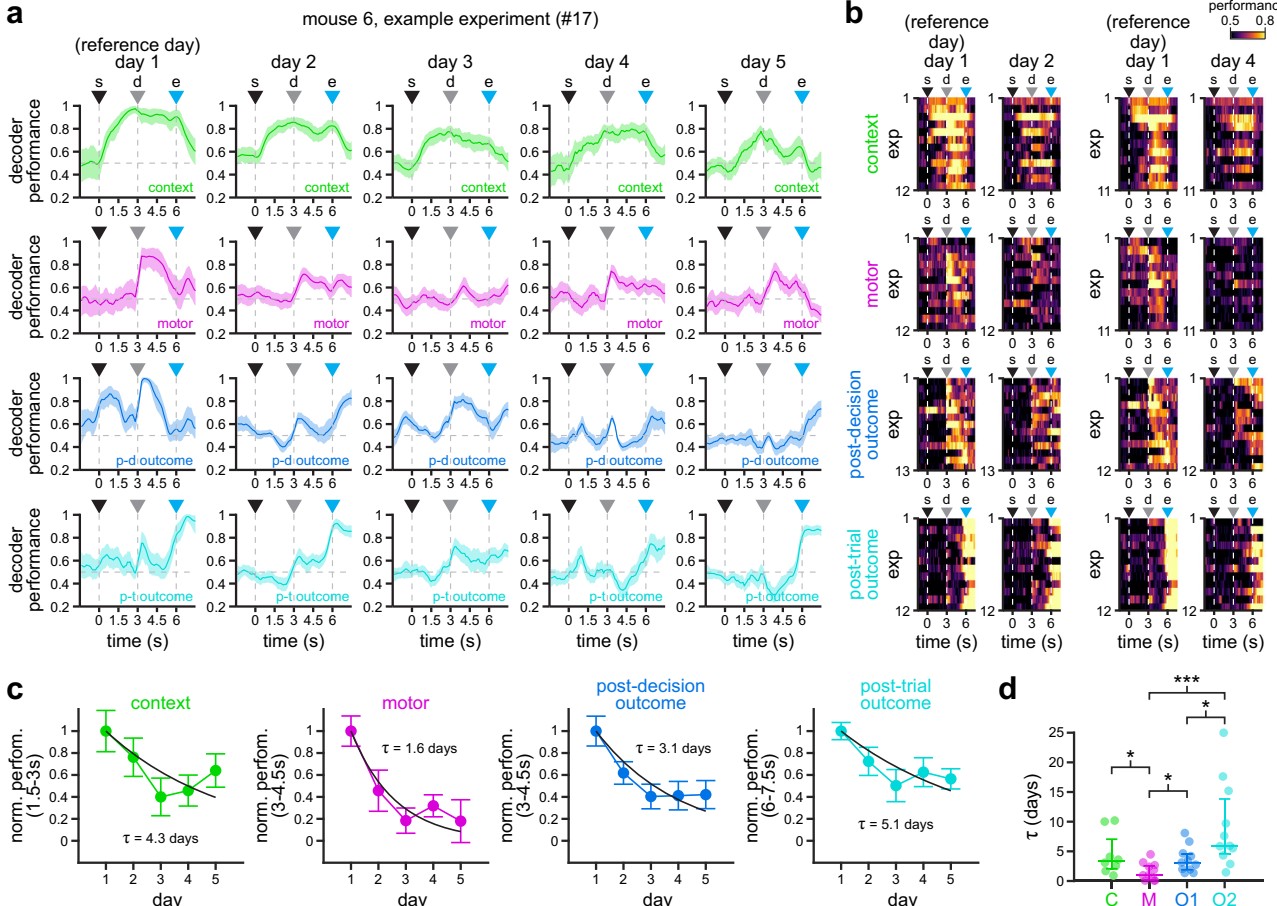

**Fig. 6 | Subpopulations of RSC neurons recorded in individual experiments exhibit different stability of task-related representations. a** Encoding of context (top row), motor choice (top-middle row), post-decision outcome (bottom-middle row), and post-trial outcome (bottom row) by populations of neurons recorded in individual experiments. Solid line, mean; shaded area, 95% CIs. Note the gradient in population encoding, where post-trial outcome and context are more stable than motor choice and post-decision outcome. **b** Heat maps of context (top), motor choice (top-middle), post-decision outcome (bottom-middle), and post-trial outcome (bottom) coding performance along trial duration for the different experiments in this study. For clarity, only comparisons between day 1 and days 2 and 4 are shown. Note the higher stability of context (top row) and post-trial outcome (bottom row) encoding in days 2 and 4 between trial start and trial end, and after trial end, respectively. By contrast, motor choice and post-decision outcome exhibit lower stability. **c** Normalized decoding performance (averaged across experiments) integrated over the indicated 1.5 s windows relevant for each task variable (mean ± s.e.m.; $n = 15, 12, 12, 11, 13$ experiments in days 1–5 [context]; $n = 16$,

12, 14, 11, 14 experiments in days 1–5 [motor]; $n = 17, 13, 14, 12, 14$ experiments in days 1–5 [post-decision outcome]; $n = 17, 12, 14, 12, 14$ experiments in days 1–5 [post-trial outcome]). Exponential decay functions were fit to estimate the decay in encoding stability across days (black lines). Note the faster decay in motor choice and post-decision outcome encoding, and the slower decay in context and post-trial outcome encoding. **d** Time constants (τ) calculated after fitting exponential decay functions to the decoding performance of each task variable in each individual experiment (see Supplementary Fig. 14a, b; $n = 8$ experiments [context]; $n = 11$ experiments [motor]; $n = 11$ experiments [post-decision outcome]; $n = 11$ experiments [post-trial outcome]). Dots, individual sessions; overlaid lines, median ± 75% CIs. For clarity, one outlier τ value for post-trial outcome (33.5 days) is shown at saturation (25 days). Asterisks denote significantly different decay constants (*$p < 0.05$, **$p < 0.01$, ***$p < 0.001$; two-sided F-test calculated by LME models; see "Methods" section for details). For all panels: s trial start, d decision point, e trial end. Source data for C and D are provided as a Source Data file.

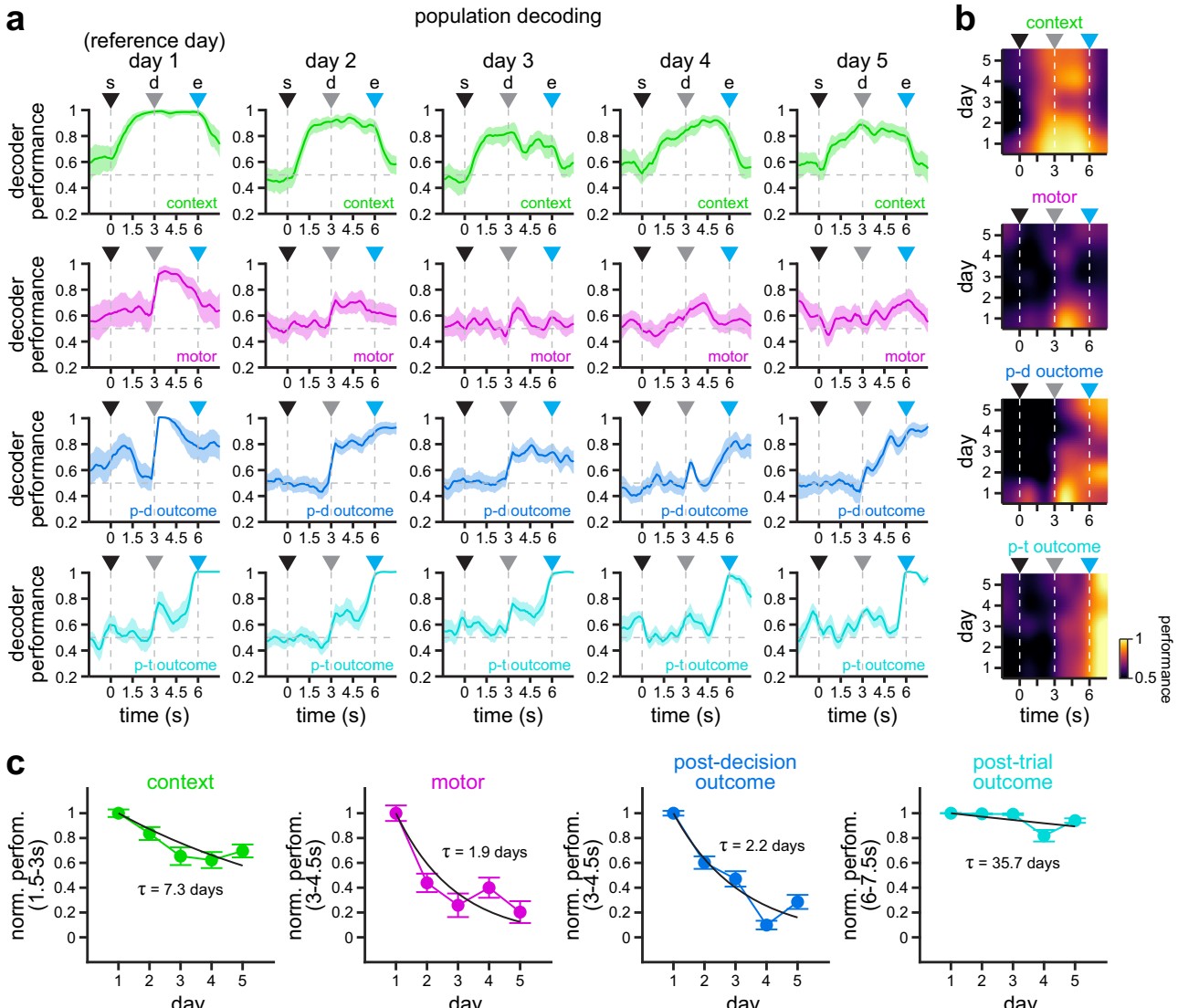

**Fig. 7 | Population-level encoding of task variables exhibits differential stability in RSC. a** Encoding of context (top row), motor choice (top-middle row), post-decision outcome (bottom-middle row) and post-trial outcome (bottom row) by pseudo-populations of RSC neurons pooled across all mice (the size of the corresponding subpopulations of neurons is indicated in Fig. 4c). Solid line, mean; shaded area, 95% CIs. Note the gradient in population encoding, where post-trial outcome and context are more stable than post-decision outcome and motor choice. **b** Heat maps of context (top), motor choice (top-middle), post-decision outcome (bottom-middle), and post-trial outcome (bottom) coding performance along trial duration and across all 5 days of experimentation. Note the stability of context encoding between trial start and trial end. Also, note the stability of post-trial outcome encoding after trial end. By contrast, the encoding of motor choice

and post-decision outcome around the decision point decays more rapidly after day 2. **c** Normalized decoding performance integrated over the indicated 1.5 s windows relevant for each task variable (mean ± bootstrapped s.e.m.; $n = 100$ iterations). Exponential decay functions were fit to estimate the decay in encoding stability across days (black lines denote fits to the average normalized performance). Note the faster decay in motor choice and post-decision outcome encoding, and the slower decay in context and post-trial outcome encoding. To obtain CIs for the decay constants, we also fit decay functions to each iteration of the decoder performance (95% CIs, context = 5.9–9.7 days, motor = 1.4–2.5 days, post-decision outcome = 1.9–2.5 days, post-trial outcome = 25.9–50.0 days). For all panels: s trial start, d decision point, e trial end. Source data for C are provided as a Source Data file.

signals to appropriate choices in the continuously changing situations that animals encounter in the environment.

## Discussion

It has been proposed that a principal function of the mammalian cortex is to preserve memories through stable representations of sensory, motor, and cognitive variables[31–33]. However, recent technological developments for longitudinal recording of neural activity have shown that neuronal response tuning can be dynamic. Often, different subsets of neurons encode information about the same cognitive variable on different days. Yet, these representations must remain

stable in order to generate reproducible behaviors over time, particularly in cortical circuits critical for performing cognitive tasks. In this study, we set out to investigate how neurons in the RSC, important for the association of contextual sensory information and motor choice[24,34], maintain their coding properties over consecutive days. We find that different task variables are represented with distinct levels of stability in the RSC, both at the population and at the single-cell levels. Environmental context, which likely reflects visual sensory features, as well as post-trial outcome are the two variables represented with the highest stability. By contrast, motor choice is less stable, likely favoring flexibility during the decision process. We argue that the

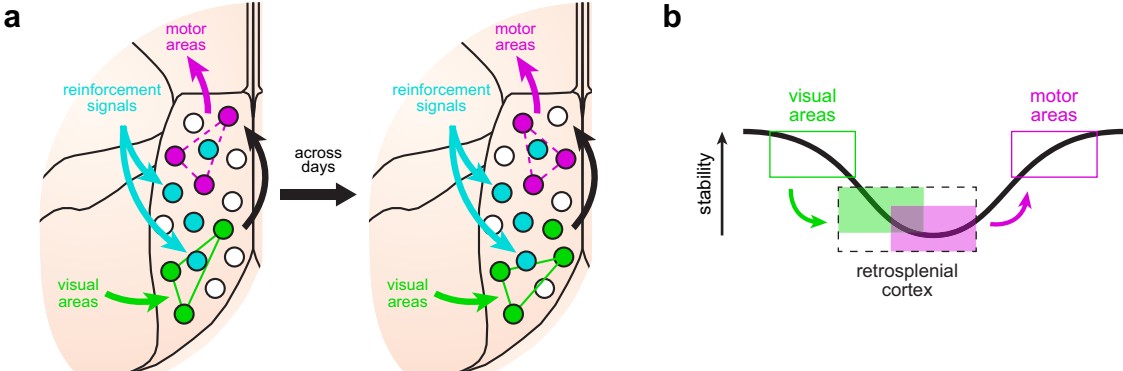

**Fig. 8 | Working model of RSC processing and representational stability. a** We hypothesize that current visual sensory features are matched to stored reference features (green arrow), which then triggers a motor output (magenta arrow) from a collection of possible choices during the association process (black arrow). Our model also postulates that sensory features are more stably represented in the RSC (solid green lines), which is supported by strong connectivity with primary and secondary visual areas[24,27,59], and that motor choice is less stable (dashed magenta lines), possibly favoring flexibility during the decision process. In addition, reinforcement signals (cyan arrows) help maintain stable responses of RSC neurons tuned to task features, particularly those related to trial outcome and

environmental context. **b** Previous studies have reported higher levels of stability in sensory and motor cortices[5,6,9,60]. Our results in retrosplenial cortex suggest that stability in association areas, which typically comprise neurons multiplexing different streams of information, follows a different function. First, sensory inputs are represented with lower stability than in the region they originate from. Motor coding, although stable within a session, changes considerably between days, which may lead to more flexible remapping to new contingencies. We expect that motor representations will become very stable again in more peripheral motor cortices.

representation of cognitive variables with different stability might be a common feature of association areas.

One key finding in our study is the differential stability of correct and incorrect responses during task performance. A possible confound is that activity levels during correct trials might be higher than in incorrect trials, thus leading to lower stability in incorrect trials consisting mainly of noise-level fluctuations with no noticeable responses. However, we did not find differences in averaged activity levels between correct and incorrect responses, except at the end of the trial, when mice received rewards only on correct trials. Hence, the observed differential stability is likely due to more consistent and reliable responses in correct trials, both within and across days. Moreover, reward may contribute to the stability of responses tuned to correct trials by enhancing memory consolidation[35,36]. Behaviorally, reduced attention and engagement in task performance might explain the higher variability and lower stability of incorrect responses across days, which might be dominated by task-independent variables in mice[37–43]. In addition, expectation, particularly after a streak of correct decisions in a single context, might modulate behavior and its underlying neural activity, biasing representations to encode previously rewarded choices[44,45], providing another substrate for increased variability in incorrect trials.

Several studies have reported that dedicated sensory or motor cortices typically encode information in a stable manner over days and weeks[5,6,9,10,14]. However, the stability of sensory and motor representations in association areas, which comprise much of the human neocortex, is less understood. Our results indicate that RSC neurons exhibit more stable responses to the environmental context than to motor choices. We postulate that the functional connectivity between context-encoding neurons and motor-encoding neurons within the RSC, which likely mediate the association process, are weak and flexible across days, while staying robust within a single day. Indeed, on individual days both context and motor choice are accurately encoded with distinct dynamics, reflecting the duration of the sensory stimulus and the window for decision, respectively, as we show here and in prior work[24].

It is possible that RSC neurons store reference sensory features that are rapidly matched to incoming sensory inputs, eliciting relatively strong context-dependent responses. However, motor planning responses might be shaped associatively on a daily basis, taking

advantage of synaptic plasticity to generate a stable motor output across days, as has been proposed for posterior parietal cortex in similar tasks[46]. In addition, motor areas might help consolidate the encoding of task features via a feedback mechanism, involving other brain regions such as the dorsal striatum[47]. Together, this evidence suggests that stable representations of sensory features in the RSC may improve the comparison of incoming inputs to stored reference contexts. At the same time, this circuit encodes higher-order motor choice outputs more dynamically, which allows updating according to task demands and provides flexibility in decision making. This is particularly relevant during navigation, which is dependent on RSC[23,24,48–56], where motor decisions must be flexibly mapped, even within a particular context. For example, the same intersection may prompt a left turn or a right turn depending on the goal location. In fact, RSC populations were recently shown to play a role in context-value associations and updating, increasing in proportion with task demands, and thus supporting cognitive flexibility across learning[25].

The implications of stable encoding of trial outcome require further consideration. One possible explanation is that reinforcement signals might help to consolidate neural activity and maintain neuronal tuning to task variables[35,36]. In this regard, a recent study found that RSC neurons have been shown to encode contexts more efficiently when they are paired with a reward[25]. Moreover, the observed stability of outcome representations, combined with stable sensory representations, might help shape motor coding on specific days, given the absence of stable motor coding across days. However, it is important to consider that the reward obtained at the end of the trial might cause lick-induced neural activity in correct trials in the RSC. This makes the stable coding at the end of the trial difficult to interpret. Nonetheless, trial outcome is also encoded right after a decision has been made, and well prior to a reward, which might reflect higher order signals related to the expectation (or the lack thereof) of a reward. Alternatively, the encoding of trial outcome during the post-decision period could represent sensory signals (such as visual or auditory) associated with the route leading to a particular outcome.

Recent findings indicate that encoding of learned cognitive variables is dynamic, exhibiting different levels of representational stability[6,17,19]. Yet, neural circuits must perform computations reliably over long periods of time to provide a substrate for robust behavior.

In this regard, it has become clear that multiplexed coding of task variables within neural circuits underlies many task-relevant transformations[22,27,38,51–54]. By balancing the differential representational stability of different cognitive variables, animals can maintain stable representations of particular variables, such as the current environmental context, while maintaining flexible associations, such as the appropriate action within that environment.

## Methods

### Animals
To achieve widespread calcium indicator expression across the dorsal cortex, we bred Emx1-Cre (Jax Stock #005628) × ROSA-LNL-tTA (Jax Stock #011008) × TITL-GCaMP6s (Jax Stock #024104) triple transgenic mice to express GCaMP6s in cortical excitatory neurons. 12–16-week-old mice of both sexes were implanted with a head plate and cranial window ($n = 6$). Water restriction started 7 days after recovery from surgical procedures, and behavioral training started after 7 days of water restriction (14 days after surgery). The animals were housed on a 12 h light/12 h dark cycle in cages of up to five animals before the implants, and individually after the implants. All animal procedures were approved by the Institutional Animal Care and Use Committee at the University of California, Santa Barbara.

### Surgical procedures
Surgeries were conducted under isoflurane anesthesia (3.5% induction, 1.5–2.5% maintenance). Deep anesthesia was monitored throughout the procedure every 15 min by confirming that respiratory rate was kept within 54–80 beats/min, body temperature within $36 \pm 1\,°C$, and there was no response to a toe pinch. Before incision, the scalp was infiltrated with lidocaine (5 mg/kg, subcutaneous) for analgesia. Meloxicam (1 mg/kg, intraperitoneal) was administered preoperatively to reduce inflammation. Once anesthetized, the scalp overlying the dorsal skull was sanitized and removed, and the periosteum was removed with a scalpel. The skull was abraded with a drill burr to improve adhesion of dental acrylic. Then, a 4–5 mm diameter craniotomy was made over the midline (centered at 2.5 to 3.0 mm posterior to bregma), leaving the dura intact. A cranial window was implanted over the craniotomy and sealed first with silicon elastomer (Kwik-Sil, World Precision Instruments) and then with dental acrylic (C&B Metabond, Parkell) mixed with black ink to reduce light transmission. The cranial windows were made of two rounded pieces of coverglass (Warner Instruments) bonded with an ultraviolet-cured optical adhesive (Norland, NOA61). The bottom coverglass (4 mm) fit tightly inside the craniotomy, while the top coverglass (5 mm) was bonded to the skull using dental acrylic. A custom-designed stainless steel head plate (eMachineShop.com) was then affixed using dental acrylic. After surgery, mice were administered carprofen (5 to 10 mg/kg, oral) every 24 h for 3 days to reduce inflammation.

### Virtual T-maze design
Mice were head-fixed using custom restraint hardware (https://goard.mcdb.ucsb.edu/resources) and placed in polypropylene tubes to limit movement. A custom rotatory joystick, located within reach of mouse forelimbs, was mounted on optical hardware (Thorlabs) and attached to an optical encoder (Digi-key Electronics). Two servomotors (Adafruit Industries) were used to constrict the position of the joystick throughout the trial, allowing movement only during the decision window. A spout was placed near to the mouth of the mouse for delivery of water rewards (10 to 12 µl), which were controlled by a solenoid valve (Parker Hannifin). Licks were detected through a capacitive touch sensor connected to the metallic spout and to a metallic mesh inside the polypropylene tube. A virtual T-maze was displayed across three screens arranged in an arc to subtend 180° of the mouse visual field. All electronic components were controlled by custom code written in MATLAB (MathWorks) through Arduino Uno (Arduino).

Virtual mazes were built using the ViRMEn package[57], with modifications to control progression through the maze. We created two different virtual T-mazes with unique wall patterns, one consisting of horizontal yellow dashes on a black background (context 1; luminance: $10.6 \pm 0.5$ lux, mean ± s.d.; root mean square contrast: 0.496) and one with oblique blue stripes on a white background (context 2; luminance: $13.4 \pm 1.0$ lux, mean ± s.d.; root mean square contrast: 0.414). The mazes were presented in pseudorandom order across an experimental session. Progression through the virtual environment proceeded at a fixed speed throughout the trial, such that both traversal from the maze start to the decision point and traversal from the decision point to the end of the maze lasted 3 s each. During the decision window, two servomotors opened to allow rotation of the joystick with the forelimbs. All mice used both forelimbs to rotate the joystick. Rotation of the joystick was coupled to rotation of the field of view in the virtual environment. If the field of view rotation exceeded 45° to the left or to the right from the central position, a decision was registered, and the joystick was returned to the central position using the servomotors. If a decision was registered, then a tone was played, indicating the trial outcome (correct: 5 kHz, 70 dB tone for 0.25 s; incorrect: broadband white noise, 80 dB for 0.25 s), and the mouse was automatically rotated 90° in the chosen direction and then progressed to the end of the selected T-maze arm. The end of the rewarded arm contained a reward cue (a blue sphere) that was reached at the same time as water was administered, whereas the unrewarded arm did not contain the reward cue. As the mouse reached the end of the maze, the outcome tone was played again at the same frequency and volume but with longer duration (correct: 5 kHz, 70 dB tone for 1 s; incorrect: broadband white noise, 80 dB for 2 s). If no decision was detected within 3 s, then the joystick was returned to the center position using the servos, and the trial was aborted and not included in later analyses. At the end of each trial, there was a 3 s interval with a uniform blue (correct trial) or black (incorrect trial) screen before teleporting back to the start of the T-maze for the next trial. To avoid repeated presentations of the same maze, the probability for consecutively displaying the same context progressively decreased after a particular context was displayed, from 50% to 0% over 10 trials. To prevent bias, if the number responses to the left or to the right exceeded double the number of responses in the nonpreferred direction over a 10 trial window, then the context rewarded for the nonpreferred direction was consecutively displayed until bias was reduced below threshold.

### Behavioral training
Before training, mice were gradually water restricted beginning 7 days after surgery. HydroGel (99% H2O, ClearH20) was provided in decreasing amounts each day for 7 days (2.0 to 1.2 g). During training, mice were supplemented with HydroGel depending on the amount of water obtained in each particular session to keep their body weight at or above 85% of their initial weight, typically 1.2 mL of water (or HydroGel equivalent) per day. In addition, once per week (before nontraining days), mice received 2.0 g of HydroGel.

Starting 7 days after water restriction, mice spent 5 days of training on a habituation and shaping protocol. During this time, we unilaterally opened the servo to only allow correct decisions (forced choice trials). This stage was important for training mice to move the joystick in both directions without bias while also instructing them on the correct context-reward associations. Following shaping, the percentage of trials with free choices was progressively increased from 10 to 80%. Only free choices were used for all further analyses. Typically, it took mice 25 to 40 sessions to achieve plateau levels of performance. Mice were trained once a day, 5–6 days a week. A single training session lasted 30 min, divided in 2 blocks of 15 min each.

### Two-photon imaging
GCaMP6s fluorescence was imaged using a Prairie Investigator two-photon microscopy system with a resonant galvo scanning module

(Bruker). For fluorescence excitation, we used a Ti:Sapphire laser (Mai-Tai eHP, Newport) with dispersion compensation (Deep See, Newport) tuned to λ = 920 nm. For collection, we used GaAsP photomultiplier tubes (Hamamatsu). To achieve a wide field of view, we used a 16×/0.8–numerical aperture microscope objective (Nikon) at an optical zoom of 2× (425 × 425 μm field). Imaging planes at a depth of 90 to 150 μm were imaged at a frame rate of 10 Hz. Laser power ranged from 40 to 75 mW at the sample depending on GCaMP6s expression levels. Photobleaching was minimal (<1%/min) for all laser powers used. A custom stainless steel light blocker (eMachineShop) was mounted to the head plate and interlocked with a tube around the objective to prevent light from the visual stimulus monitor from reaching the photomultiplier tubes.

To achieve longitudinal imaging across days, we identified the same field of view using the blood vessel pattern as a guide, and by measuring the depth from the surface of the cortex with micrometric precision. Landmark cells and/or small blood vessels were used for fine adjustments to match the field of view.

## Two-photon postprocessing

Images were acquired using PrairieView acquisition software and converted into TIF files. All subsequent analyses were performed in MATLAB (MathWorks) using custom code (https://goard.mcdb.ucsb.edu/resources). First, images from a single 15 min recording block (2 blocks per 30 min experimental session) were corrected for X-Y movement by rigid registration to a reference image (the pixel-wise mean of all frames in the middle third of the recording block) using two-dimensional cross-correlation. Then, each block (2 per day) was independently registered to the first block of the first day of experimentation using the two-dimensional cross-correlation of their average projections. This yielded a 150 min long stack containing images across a 5-day experiment. To identify responsive neural somata, a pixel-wise activity map was calculated over each 150 min long stack using a modified kurtosis measure. Neuron cell bodies were identified using local adaptive threshold and iterative segmentation. Automatically defined regions of interest (ROIs) were then manually checked for proper segmentation in a graphical user interface (GUI) that allows comparison between raw fluorescence and activity map images. The corresponding GUI for registration of video stacks and refinement of ROIs can be found at: https://github.com/ucsb-goard-lab/defineCellROIs.

To ensure that the response of individual neurons was not due to local neuropil contamination of somatic signals, a corrected fluorescence measure was estimated according to

$$F_{corrected}(n) = F_{soma}(n) - \alpha*\left(F_{neuropil}(n) - F_{neuropil}\right) \quad (1)$$

where $F_{neuropil}$ was defined as the fluorescence in the region within 30 μm from the ROI border (excluding other ROIs) for frame $n$, $F_{neuropil}$ is the average neuropil fluorescence across the entire time series, and $\alpha$ was chosen from [0 1] for each neuron to minimize the Pearson's correlation coefficient between $F_{corrected}$ and $F_{neuropil}$. The $\Delta F/F(\%)$ for each neuron was then calculated as:

$$\Delta F/F(\%) = 100*\left(F_n - F_0\right)/F_0 \quad (2)$$

where $F_n$ is the corrected fluorescence ($F_{corrected}$) for frame $n$, and $F_0$ is defined as the mode of the corrected fluorescence density distribution across the entire time series.

For some analyses (Supplementary Fig. 18), we estimated neuronal spikes from calcium signals using a MATLAB implementation of a sparse, nonnegative convolution algorithm (OASIS) on $\Delta F/F$ traces with an autoregressive model of order 1 for the convolution kernel[58].

## Analysis of two-photon imaging data

To determine whether a neuron exhibited consistent task-related activity, we calculated the Pearson's correlation coefficient between odd and even correct trials and compared it to the correlation coefficient of trials in which activity was circularly permuted at a random point on each trial (time shuffled). If the original correlation coefficient was greater than the 95th percentile of the correlation coefficients from the time-shuffled activity (100 iterations), then a cell was considered to exhibit reliable task-related activity for that specific day. Only cells with reliable activity in day 1 and in at least one of days 2 to 5 were used for specific day-to-day comparisons. Furthermore, we evaluated whether each cell ROI was morphologically stable across experimental sessions (Supplementary Fig 3a). To achieve this, we centered each ROI within a 30 × 30 μm window, and calculated the structural similarity index (SSI; Matlab ssim function) between the windows in day 1 and days 2 to 5. We then compared each SSI to a null distribution of SSIs between the corresponding window in day 1 and random windows in other days. If the true SSI value was higher than the 95th CI of the SSI null distribution, then an ROI was considered stable for that specific day-to-day comparison (Supplementary Fig. 3b). We also checked using the 2-D correlation coefficient as a metric of similarity between windows containing ROIs, obtaining similar results (Supplementary Fig. 3c). In addition to activity reliability and ROI morphological stability, each day-to-day comparison had to meet behavioral criteria: performance above chance level (>0.5), and ≥2 trials of each trial type (correct-left, incorrect-left, incorrect-right, correct-right; Supplementary Fig. 1).

Trained mice typically make more correct than incorrect behavioral responses. Thus, in order to achieve an unbiased comparison of the neuronal activity in correct and incorrect trials, we resampled trials without replacement (100 iterations) to the lowest number of trials in each context for each neuron. In most cases, this resulted in neural responses resampled to the number of incorrect trials, although this was not always the case (see Supplementary Fig. 1a).

Preference for a task variable was determined by resampling trials with replacement (100 iterations) and grouping accordingly. For instance, to evaluate context preference, correct left and incorrect right trials (context 1) are resampled and grouped together, and compared to resampled incorrect left and correct right trials (context 2). Then, if the activity is significantly different (Wilcoxon signed-rank test) for 5 consecutive bins between the two populations of resampled trials, a cell is considered to exhibit preference for the trials with higher levels of activity. In addition, to evaluate differences in task variable preference, the subpopulations of cells with preference for a particular task variable were resampled for each day (100 iterations). Preference was considered different when the 95% CIs of their bootstrapped distribution did not overlap (Fig. 3b).

To measure the stability of responses in individual cells, we calculated the Pearson's correlation coefficient ($CC$) between odd trials on day 1 and even trials on days 1–5. This was done either directly, for each trial type separately (correct left, incorrect right, incorrect left, and correct right; Fig. 3d), or by grouping activity by their preferred task variable (context 1/context 2, left/right choice, or correct/incorrect outcome; Fig. 3c). Next, to quantify the stability of the $CC$ across days, we fit exponential decay functions to each individual cell:

$$CC = CC_{day1}*e^{\frac{1-day}{\tau}} \quad (3)$$

to obtain their fit parameters $CC_{day1}$ and $\tau$. However, since fit parameters were not normally distributed, we first calculated their rankit:

$$\text{fit parameter} = \text{invnormal}\left(\frac{\text{rank of fit parameter} - 0.5}{n}\right) \quad (4)$$

and then used LME models to determine significant differences among groups.

The coding performance of task variables was determined by support vector machine classifiers[24]. To achieve this, trials were homogenized by sampling an equal number of each trial type with replacement (correct left, incorrect right, incorrect left, and correct right), and then splitting them into two nonoverlapping groups containing 50% trials of each type. This process ensures that all conditions comprise the same number of trials. For analyses within a single day, a support vector machine classifier was trained using 50% of trials and tested on the remaining 50% of trials per time bin. For analyses across days, and to maintain similar sampling, 50% of randomly sampled trials were used for training the classifiers in reference day 1, and 50% of randomly sampled trials were used for testing these classifiers in days 2 to 5. The performance confidence intervals (CI) were obtained by calculating the 5th and 95th percentiles of the bootstrapped data (100 iterations). The encoding of the different variables was determined by comparing conditions as follows: (1) context, correct left and incorrect right versus incorrect left and correct right; (2) motor, correct left and incorrect left versus incorrect right and correct right; and (3) outcome, correct left and correct right versus incorrect right and incorrect left. A cell was considered to encode any of the task variables if the performance of the decoder was higher than chance for at least 5 consecutive time bins (lower 5th CI of the bootstrapped distribution > 0.5; see Supplementary Fig. 8).

To measure the stability of decoding performance (P) in individual cells (Figs. 4c and 5c, and Supplementary Fig. 12c), we fit exponential decay functions to each cell:

$$P = P_{day1} * e^{\frac{1-day}{\tau}} + 0.5 \qquad (5)$$

This allowed us to compare the fit parameters $P_{day1}$ and $\tau$ among groups using LME models on their corresponding rankit values. The constant 0.5 is added to account for the baseline where decoding performance reaches chance level. In Fig. 4c, the top left inset for $P_{day1}$ takes this 0.5 into account.

For some analyses of decoding stability, we integrated between chance level (0.5) and performance level for time windows relevant for each task variable (1.5 s), and normalized by the integrated performance in day 1 (when the classifier is trained). We then fit an exponential decay function to this normalized performance (NP):

$$NP = e^{\frac{1-day}{\tau}} \qquad (6)$$

to estimate the time at which decoding over chance level decays to 1/e (~37%) of its original value in day 1. Note that NP values = 1 indicate a decoding performance equal to that in day 1, and that NP values = 0 signify decoding performance equal to chance level. Thus, NP is a metric that allows us to evaluate the stability in the decoding of task variables across days. In addition, to obtain CIs for the decay constants, we also fit decay functions to each iteration of the classifier performance. This exponential decay function generally fit the decay in decoding performance well (Figs. 6c and 7c and Supplementary Figs. 14c, 16c, 17c, and 18d), particularly when enough data was available to calculate the decoding performance of task variables. However, for one analysis, we estimated decay constants for each individual experiment or each individual mouse (Fig. 6d and Supplementary Fig. 14d), which, in some cases, resulted in poor fits. Thus, we only considered fits with low sum of squares (SSE) values (≤ 1.6, preserving 87.9% of fits; based on the distribution of SSE values), and positive R-squared values (fits are better than a horizontal line through the mean value). Then, the rankit of their corresponding decay constants were compared using LME models. In addition, we used a slightly different model for the fits on not normalized data (Supplementary Fig. 19), adding one term for the baseline = 0.5, and one term for the performance in day 1:

$$P = P_{day1} * e^{\frac{1-day}{\tau}} + 0.5 \qquad (7)$$

In Supplementary Fig. 19, the value for $P_{day1}$ takes this 0.5 into account.

## Statistical information
When possible, data groups are compared using linear mixed-effects (LME) models using task variables or trial type for the fixed effect for the intercept, and the experiment number and the mouse number as nominal values for the random effects for the intercept. All three terms are modeled as predictor variables of the particular data being compared (e.g., correlation coefficient, classifier performance). We fit the parameters of mixed-effect models using the restricted maximum likelihood (REML) estimation, which estimates the variance parameters independently of the estimates for the fixed effects. When independent sampling was not possible (e.g., population vector correlation coefficients, population decoder performance), data were compared using bootstrap estimates via sampling with replacement (100 iterations), and differences were determined by nonoverlap of the 95% CIs. Bootstrap estimates of s.e.m. were calculated as the s.d. of values evaluated in shuffled iterations. Very small p values (<$10^{-5}$) were capped at $p \leq 10^{-5}$ as a lower bound on reasonable probabilities.

## Reporting summary
Further information on research design is available in the Nature Portfolio Reporting Summary linked to this article.

## Data availability
The neural activity data and the task variable decoding data generated in this study have been deposited in the Dryad database under a CC0 1.0 Universal (CC0 1.0) Public Domain Dedication license (https://doi.org/10.5061/dryad.7pvmcvf2z). In addition, the processed data related to all statistical analyses in the main figures are available in the Source Data file. Source data are provided with this paper.

## Code availability
The code for automated detection and refinement of ROIs across imaging sessions is available at this Github repository: https://github.com/ucsb-goard-lab/defineCellROIs.

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

## Acknowledgements

We would like to thank Ralf Wessel, Tina Xia, and Jhoseph Shin for their comments on the manuscript and code review. This work was supported by the following: Harvey Karp Discovery Award (L.M.F.), UC MEXUS-CONACYT Postdoctoral Fellowship (L.M.F.), NIH R01 NS121919 (M.J.G.), NSF 1707287 (M.J.G.), Larry L. Hillblom Foundation (M.J.G.), and Whitehall Foundation (M.J.G.).

## Author contributions

L.M.F. and M.J.G. designed the experiments; L.M.F. performed the surgical implants; L.M.F. conducted the experiments and analyzed the data; L.M.F. and M.J.G. wrote the manuscript.

## Competing interests

The authors declare no competing interests.
