## [Peer Review File · Nature Communications]

Reviewers' comments:

Reviewer #1 (Remarks to the Author):

Using longitudinal two-photon microscopy, the authors measured calcium responses in thousands of RSC neurons across five consecutive days in a context-motor choice association task. They claim that RSC representations of context and trial outcome display higher stability than motor choice. The question they address is of significant interest but some aspects of the analysis need to be improved or clarified.

Stability was defined as “cells with at least 5 bins with significant decoding that were also significant on day 1”. This looks arbitrary, as one can also integrate performance over all bins, or take the maximal or the mean value at task-related time intervals. More efforts are necessary to justify the choice made. It seems that the outcome of stability analysis may substantially depend on which metric is used.

When “Response correlation” is used in 2C for analysis of stability in neuronal activity, the declines look very similar (should be estimated and given as in 5C) for all task-related measures, only the basal variability related to trial outcome is lower on day 1. Also for the fraction of cells with stable encoding in Fig. 3E, the major difference is between the trial outcome and two other measures. This may be related to the dependence of neuronal activity on licking, which starts close to the end of the trial when a mouse is expecting the reward. This factor should be taken into account in the analysis e.g. by excluding intervals related to licks.

When AUC is used as a measure of stability in 5C, it provides an outcome very different from the two other mentioned measures, with context representation as the most stable. However, this may just reflect the fact that 400 neurons are used for the prediction of context, while only about 100 neurons are used for the prediction of two other measures. This analysis should be redone using the same number of cells.

It seems that AUCs for context/outcome/motor choice were normalized to one on day 1. This should be mentioned in the methods and/or figure legend. It could be interesting to see not-normalized curves and eventually to relate the decay rate to AUC values on day 1, as a measure related to the initial “quality” of representation.

The functional relevance of the used stability measures is unclear. In the concluding sentence of the abstract, the authors relate it to “the flexibility to subserve behavioral demands” but this is not

shown. One option would be to study the stability of AUC for individual animals and relate the loss of stability in representations of each of three task-related measures on day X+1 as compared to day X to the success rate on day X+1.

It could be essential to show AUCs for optimal classifiers with parameters determined on each of the studied days. It could be that instability reflects fine-tuning/improvement of representations.

In any case, to understand how robust are the conclusions of this study (mostly based on data from 4 animals), the comparison of stability curves for task-related measures should be also done using an animal rather than a neuron as an observation unit.

In the Abstract, the motor choice is not well introduced, it may be better to refer to “a context-motor choice association task”.

Locations corresponding to s,d,e could be shown in Fig. 1A.

The authors stated in the Introduction “However, it is not known how stable sensory and motor variables are represented in RSC during the performance of a cognitive task”. However, Sun et al. (2021) Nat Commun studied cell remapping of cell types during context discrimination and reversal learning tasks and although the stability was not the major aspect of that work and a different measure of stability was used, it may be worth discussing it in the context of the present study.

Page 4: ...rewards (Fig. 1A; for a full description of this task, see Methods and ref. 22...

The closing bracket is missing.

Fig1.D vs E Different colormaps are used: on days 2-5 correlations are lower. Was R or R² used – why there are no negative correlations?

Page 4: To ensure this was not due to unequal trial numbers, we sampled equally across trial types (Fig. S2A), and confirmed that population vectors in correct trials exhibit higher stability (Fig. 2B-E).

Fig2 has no panels D,E

Fig2B: Significance is missing.

Legend to Figure 3E: Asterisks denote significantly higher stability. Significance was defined as a lower 5% CI higher than higher 95% CI among task variables.

However, a lack of 95%CI overlap does not mean significance. If one wants to test H0 that the observed parameter difference is not random, one must permute the groups. In this case it may be much simpler and sufficient to use GLME for binomial distribution.

Figure 5C, page 15 (lines 1-5): No statistical analysis was performed.

Page 23. Two-photon postprocessing: We greatly appreciate that the authors added a link to the GitHub repository. But it is not of optimal quality. Moreover, without a data example, these scripts don't work. It should be crucial to document the quality of multi-day registration of studied neurons. This quality is the key point of this article. Unfortunately, this issue is not well discussed. It is impossible to test the code given on GitHub, as there is no test data, so please consider to add.

Page 24. Analysis of two-photon imaging data: "To determine whether a neuron exhibited consistent task-related activity, we calculated the Pearson's correlation coefficient between odd and even correct trials and compared it to the correlation coefficient of trials in which activity was circularly permuted at a random point on

each trial (time shuffled). If the original correlation coefficient was greater than the 97.5th percentile of the correlation coefficients from the time shuffled activity (1000 iterations), then the neuron was considered to exhibit reliable task-related activity and used for all subsequent analyses."

As calcium signals may have temporal structure, the H0 distribution should not be obtained by the permutation of the signal, but by calculating the correlation between random trials obtained from any neurons of a given mouse.

Reviewer #2 (Remarks to the Author):

Reviewer Comments

This study investigated stability of representations of behaviorally relevant variables in association areas in retrosplenial cortex using 2P calcium imaging. The study suggests that neural activity

exhibits different levels of stability across days. They use a virtual T-Maze task with 2 different contexts for mice to turn left or right and 2P imaging in Layer 2/3 of RSC.

They state specifically RSC representations on context and trial outcome are more stable than motor choice. As an interpretation of these findings, authors suggest diverse information is stored with varying levels of stability, balancing stability and flexibility.

Major Concerns:

1. I did not feel the study and the paper is thorough. The study and figures do not present key elements of data

A. Not a single field of view or raw activity trace is shown. Please provide

B. No quantification of stability of the field of view across days (see below)

C. Are all ROIs pooled across animals? Please show individual animal data and compare stats across animals

2. I have concerns with the way the data is processed and analyzed. Unlike standard methods in the field no transient detection was done, so all correlations are done on continuous traces. The disadvantage to this is that PV overlap could be affected by different brightness levels in different parts of the field of view

3. I was confused with some aspects of figure 2. Cell 1 is described as exhibiting higher responses for context 1, but the responses seem to be clearly higher for context 2. Cell 2 is described as exhibiting higher responses for right turns, but this appears to be the case only for correct trials, though the effect is visible in some incorrect trials.

4. In Fig 3, are these cells pooled from several mice? Are there sufficient N to show similar patterns across mice?

5. In 5C, there is an additional distinction between post-decision outcome and post-trial outcome. Which of those is analogous to outcome in the other figures?

Related, figure 5C possibly contradicts figure 3E. From 3E, I would expect outcome to be far more stable than context and motor, but in 5C, context is more stable than post-decision outcome, and both are more stable than motor. Post-trial outcome is clearly the most stable, but I am unsure how to combine that with post-decision outcome.

6. It would be interesting to see if there was increased stability by a later stage of learning. All of the data presented is based on days with a handful of miss trials. Neighboring day analyses would also be helpful as a supplement. Also does 5 consecutive days seem like enough time to track stability. In the hippocampus studies track over longer time period (7-60 days). Please provide quantifiable measures of stability (PV Correlation; stability index; TC correlation, Representational Drift index)

Minor:

1. In figure 3 the example cell shown for decoding motor choice seems to be very weakly modulated. Is there another possible example that has better performance?
2. Unclear why there are fewer cells in figure 3 than indicated in Figure 2B. If the numbers in figure 3 are correct, that implies that in Figure 2B there should be many cells classified as having no preference.
3. Is there any topographical organization of cells with different response properties, as hinted at in Figure 5D? Do nearby neurons respond similarly, or is it a salt and pepper arrangement?

Reviewer #3 (Remarks to the Author):

Reviewer Comments

This study investigated stability of representations of behaviorally relevant variables in association areas in retrosplenial cortex using 2P calcium imaging. The study suggests that neural activity exhibits different levels of stability across days. They use a virtual T-Maze task with 2 different contexts for mice to turn left or right and 2P imaging in Layer 2/3 of RSC. They state specifically RSC representations on context and trial outcome are more stable than motor choice. As an interpretation of these findings, authors suggest diverse information is stored with varying levels of stability, balancing stability and flexibility.

Major Concerns:

1. I did not feel the study and the paper is thorough. The study and figures do not present key elements of data
 - A. Not a single field of view or raw activity trace is shown. Please provide
 - B. No quantification of stability of the field of view across days (see below)
 - C. Are all ROIs pooled across animals? Please show individual animal data and compare stats across animals
2. I have concerns with the way the data is processed and analyzed. Unlike standard methods in the field no transient detection was done, so all correlations are done on continuous traces. The disadvantage to this is that PV overlap could be affected by different brightness levels in different parts of the field of view
3. I was confused with some aspects of figure 2. Cell 1 is described as exhibiting higher responses for context 1, but the responses seem to be clearly higher for context 2. Cell 2 is described as exhibiting higher responses for right turns, but this appears to be the case only for correct trials, though the effect is visible in some incorrect trials.
4. In Fig 3, are these cells pooled from several mice? Are there sufficient N to show similar patterns across mice?
5. In 5C, there is an additional distinction between post-decision outcome and post-trial outcome. Which of those is analogous to outcome in the other figures? Related, figure 5C possibly contradicts figure 3E. From 3E, I would expect outcome to be far more stable than context and motor, but in 5C, context is more stable than post-decision outcome, and both are more stable than motor. Post-trial outcome is clearly the most stable, but I am unsure how to combine that with post-decision outcome.

6. It would be interesting to see if there was increased stability by a later stage of learning. All of the data presented is based on days with a handful of miss trials. Neighboring day analyses would also be helpful as a supplement. Also does 5 consecutive days seem like enough time to track stability. In the hippocampus studies track over longer time period (7-60 days). Please provide quantifiable measures of stability (PV Correlation; stability index; TC correlation, Representational Drift index)

Minor:

1. In figure 3 the example cell shown for decoding motor choice seems to be very weakly modulated. Is there another possible example that has better performance?
2. Unclear why there are fewer cells in figure 3 than indicated in Figure 2B. If the numbers in figure 3 are correct, that implies that in Figure 2B there should be many cells classified as having no preference.
3. Is there any topographical organization of cells with different response properties, as hinted at in Figure 5D? Do nearby neurons respond similarly, or is it a salt and pepper arrangement?

Response to Reviewers

We thank the reviewers for taking the time to provide detailed feedback on our manuscript. We have made a major revision to the manuscript, including much more detail on the physiology experiments, behavior, and cell tracking. We have used a new analysis approach that improves our included sample size (mice, imaging fields, and neurons). Finally, we carried out a number of control analyses suggested by the reviewers. We believe these changes have significantly improved the strength of the paper.

Reviewer #1 (Remarks to the Author):

Using longitudinal two-photon microscopy, the authors measured calcium responses in thousands of RSC neurons across five consecutive days in a context-motor choice association task. They claim that RSC representations of context and trial outcome display higher stability than motor choice. The question they address is of significant interest but some aspects of the analysis need to be improved or clarified.

Thank you for the comments, we have addressed the specific concerns with the analysis below.

Stability was defined as “cells with at least 5 bins with significant decoding that were also significant on day 1”. This looks arbitrary, as one can also integrate performance over all bins, or take the maximal or the mean value at task-related time intervals. More efforts are necessary to justify the choice made. It seems that the outcome of stability analysis may substantially depend on which metric is used.

We agree that it is important to be very clear about our stability metric. We added a new supplementary figure showing example cells that had significant or non-significant responses over five sessions (Fig S7). We defined significant responses in this manner because different neurons respond during different parts of the trial. We tried integrating over the entire trial, but we found that this metric causes us to miss neurons that are transiently—but consistently—active during a small window of the trial.

We did try the second approach the reviewer suggested—using the mean value at task-related intervals—and found very similar results (Fig R1, below).

Figure R1: Analysis of task variable stability (similar to Fig 7C) calculated using neurons that exhibit significant responses during particular task-related time intervals (1.5-3s, 3-4.5s, and 6-7.5s).

In addition, to make sure that our choice of Day 1 as the reference day did not influence the data, we carried out the analysis using each of the 5 days as reference points and found qualitatively similar results (Fig S15).

When “Response correlation” is used in 2C for analysis of stability in neuronal activity, the declines look very similar (should be estimated and given as in 5C) for all task-related measures, only the basal variability related to trial outcome is lower on day 1. Also for the fraction of cells with stable encoding in Fig. 3E, the major difference is between the trial outcome and two other measures. This may be related to the dependence of neuronal activity on licking, which starts close to the end of the trial when a mouse is expecting the reward. This factor should be taken into account in the analysis e.g. by excluding intervals related to licks.

We agree with this comment, and have separated out post-decision outcome responses and post-trial outcome responses for all analyses in Fig 3C (formerly Fig 2C) and Fig 4A-C (formerly Fig 3A-E). We decided to keep post-trial outcome measures, but we discuss the caveats in interpretation both in the main text (page 14, paragraph 1, lines 14-16) and the discussion section (page 23, paragraph 2, lines 7-14). We do still observe higher response correlation for the post-decision outcome measure than for context and motor (Fig 3C). We suspect that this metric is influenced by the structure of the signal, with transient signal exhibiting higher correlations than broad signals. This was a principal motivation for using the decoding analyses used throughout the rest of the manuscript. We discuss this on page 12, paragraph 1, lines 1-6.

When AUC is used as a measure of stability in 5C, it provides an outcome very different from the two other mentioned measures, with context representation as the most stable. However, this may just reflect the fact that 400 neurons are used for the prediction of context, while only about 100 neurons are used for the prediction of two other measures. This analysis should be redone using the same number of cells.

We repeated the analysis using the same number of neurons for each task variable (Fig S18B). The results were generally similar, with a slight decrease in stability for post-trial outcome, which had the highest percentage of significantly encoding cells.

It seems that AUCs for context/outcome/motor choice were normalized to one on day 1. This should be mentioned in the methods and/or figure legend. It could be interesting to see not-normalized curves and eventually to relate the decay rate to AUC values on day 1, as a measure related to the initial “quality” of representation.

Thank you for catching that mistake, we have corrected that in the axis labels, legends, and methods (page 30, paragraph 2, lines 1-3). We have added a plot of unnormalized curves to Fig S18A. As indicated in the figure, the Day 1 performance was similar (>90%) across variables, and not directly related to cross-day stability. For example, “post-trial outcome” responses had higher Day 1 encoding than “Context”, but lower cross-day stability.

The functional relevance of the used stability measures is unclear. In the concluding sentence of the abstract, the authors relate it to “the flexibility to subserve behavioral demands” but this is not shown. One option would be to study the stability of AUC for individual animals and relate the loss of stability in representations of each of three task-related measures on day X+1 as compared to day X to the success rate on day X+1.

This is an interesting idea, but unfortunately we found the variance in behavioral performance was pretty low, and as a result we did not observe any correlation between encoding and performance changes across days. Even if we did, it would be difficult to rule out extraneous factors affecting both coding and performance (e.g., inattention, grooming, etc).

We are very interested in the idea that some amount drift in particular features/variables may be beneficial to coding (e.g., Masset et al., 2022; Driscoll et al., 2022), and wanted to discuss our results in this context. However, we want to make it clear that there is no strong evidence for this claim as of yet, and have tempered our claims in the abstract and conclusions.

It could be essential to show AUCs for optimal classifiers with parameters determined on each of the studied days. It could be that instability reflects fine-tuning/improvement of representations.

We have done this analysis and indeed found that cross-validated performance of both individual neurons (Fig S8) and populations (Fig S14) perform quite similarly on individual days. We have discussed this more detail in the main text Discussion section (last paragraph on page 22).

In any case, to understand how robust are the conclusions of this study (mostly based on data from 4 animals), the comparison of stability curves for task-related measures should be also done using an animal rather than a neuron as an observation unit.

To address this, we made a major change to our analysis approach. In the original manuscript we only used neurons in which all five days of recording met all behavioral criteria (sufficient performance and number of error trials). We have kept this analysis as a supplementary figure (Fig S16), but for the rest of the manuscript we used all pairs of sessions in which behavioral conditions were met in both sessions. This significantly increased the number of experiments we could use, since we did not have to exclude an entire 5-day set of sessions just because one session had insufficient error trials or poor behavioral performance. This allowed us to increase our sample of mice, sessions, and neurons. With this larger sample, all of the main effects were still present, indicating the effects are quite robust across animals and experiments.

Our animal sample size ($N = 6$ mice) was still too small for robust statistical comparisons, but we were able to perform our analysis using each imaging field as a unit ($N = 8-11$ fields from 6 mice) and found very similar results (Fig 6B-D; see Fig S13 for curves from individual fields).

In the Abstract, the motor choice is not well introduced, it may be better to refer to “a context-motor choice association task”.

Thank you, fixed.

Locations corresponding to s,d,e could be shown in Fig. 1A.

We have added these location markers to Fig. 1A.

The authors stated in the Introduction “However, it is not known how stable sensory and motor variables are represented in RSC during the performance of a cognitive task”. However, Sun et al. (2021) Nat Commun studied cell remapping of cell types during context discrimination and reversal learning tasks and although the stability was not the major aspect of that work and a different measure of stability was used, it may be worth discussing it in the context of the present study.

Thank you for pointing out this article, we added discussion of it to the manuscript (page 23, paragraph 1).

Page 4: ...rewards (Fig. 1A; for a full description of this task, see Methods and ref. 22...
The closing bracket is missing.

Thank you, fixed.

Fig1.D vs E Different colormaps are used: on days 2-5 correlations are lower. Was R or R² used – why there are no negative correlations?

Thank you for catching that. Since the vast majority of correlations were between $0 < R < 0.7$, we used those as the ends of our colormap to better display the dynamic range (i.e., <0 would appear as dark blue, values >0.7 would appear as dark red). However, we did not state that in the figure legend - we have corrected that oversight.

Page 4: To ensure this was not due to unequal trial numbers, we sampled equally across trial types (Fig. S2A), and confirmed that population vectors in correct trials exhibit higher stability (Fig. 2B-E).

Fig2 has no panels D,E

Thank you, fixed.

Fig2B: Significance is missing.

We have added significance testing to the figure subpanel (now Fig. 3B).

Legend to Figure 3E: Asterisks denote significantly higher stability. Significance was defined as a lower 5% CI higher than higher 95% CI among task variables.

However, a lack of 95%CI overlap does not mean significance. If one wants to test H₀ that the observed parameter difference is not random, one must permute the groups. In this case it may be much simpler and sufficient to use GLME for binomial distribution.

We replaced this with a new analysis using linear mixed-effects models (Fig. 4C). For some analyses (e.g., decoding analyses) we use bootstrapped data, in which case we cannot use linear mixed-effects models or other statistical tests that require independence of samples - for these comparisons we use non-overlap of 95% CI as an indicator of differential coding.

We took care to re-test all principal analyses using approaches that do not require bootstrapping (e.g., Fig 6D).

Figure 5C, page 15 (lines 1-5): No statistical analysis was performed.

We have added significance testing, both using individual experiments (Fig 6D, page 16-17) and by bootstrapping with the entire population (Fig 7C, page 18-19).

Page 23. Two-photon postprocessing: We greatly appreciate that the authors added a link to the GitHub repository. But it is not of optimal quality. Moreover, without a data example, these scripts don't work. It should be crucial to document the quality of multi-day registration of studied neurons. This quality is the key point of this article. Unfortunately, this issue is not well

discussed. It is impossible to test the code given on GitHub, as there is no test data, so please consider to add.

Our apologies, we had not finished updating the GitHub repository. We changed our analysis approach to more carefully screen for well-tracked neurons using structural similarity index (Fig 1G; Fig S2A-C). We have updated the GitHub repository and included sample data: <https://github.com/ucsb-goard-lab/defineCellROIs/tree/main>

Page 24. Analysis of two-photon imaging data: “To determine whether a neuron exhibited consistent task-related activity, we calculated the Pearson’s correlation coefficient between odd and even correct trials and compared it to the correlation coefficient of trials in which activity was circularly permuted at a random point on each trial (time shuffled). If the original correlation coefficient was greater than the 97.5th percentile of the correlation coefficients from the time shuffled activity (1000 iterations), then the neuron was considered to exhibit reliable task-related activity and used for all subsequent analyses.”

As calcium signals may have temporal structure, the H0 distribution should not be obtained by the permutation of the signal, but by calculating the correlation between random trials obtained from any neurons of a given mouse.

Our apologies if we are missing something, but we did not understand this point. The reason for circular permutation the signal (instead of shuffling time points) is to preserve the temporal structure of the calcium signal for that particular neuron when generating the H0 distribution. This allows us to determine if a neuron exhibits significant task-evoked responses compared to a simulated neuron with similar activity levels, but that is not driven by the task. Using trials from other neurons (not shifted in time) would lead to high correlations in the H0 distribution since some neurons have similar responses during the task.

Reviewer #2 (Remarks to the Author):

Reviewer Comments

This study investigated stability of representations of behaviorally relevant variables in association areas in retrosplenial cortex using 2P calcium imaging. The study suggests that neural activity exhibits different levels of stability across days. They use a virtual T-Maze task with 2 different contexts for mice to turn left or right and 2P imaging in Layer 2/3 of RSC. They state specifically RSC representations on context and trial outcome are more stable than motor choice. As an interpretation of these findings, authors suggest diverse information is stored with varying levels of stability, balancing stability and flexibility.

Major Concerns:

1. I did not feel the study and the paper is thorough. The study and figures do not present key elements of data

We realize that we moved to the analysis fairly quickly in the original manuscript, so we have added several new figures on the physiology and cell tracking to address this issue.

A. Not a single field of view or raw activity trace is shown. Please provide

We added an example field, shown across five sessions (Fig. S2A). We also added DF/F traces from example cells across five sessions (Fig. 1H). In addition, to better illustrate the data collection, we added a video showing behavior and imaging across multiple days (Video S1).

B. No quantification of stability of the field of view across days (see below)

We have added more careful quantification of stability for each cell across days using the structural similarity index of average projection in the local region centered on each ROI (Fig S2; analysis loosely based on Sheintuch et al., *Cell Reports*, 2017). Although the vast majority of our neurons were consistent across sessions, we did exclude a subset (typically <10%) in which the average projection was not sufficiently stable across sessions.

C. Are all ROIs pooled across animals? Please show individual animal data and compare stats across animals

We added a new supplementary figure showing the stability curves for each animal, showing that the main effects are consistent among animals. Unfortunately, the number of animals included ($n = 6$ using the new analysis approach) was too small for robust statistical analysis. Instead, we re-analyzed the data using each imaging field separately to avoid overrepresentation of particular imaging fields (Fig. 6). We should also note that the number of cells per experiment and per mouse were relatively even (Fig. 1D), so the data are unlikely to be attributable to a small number of experiments having a strong effect on the overall data.

2. I have concerns with the way the data is processed and analyzed. Unlike standard methods in the field no transient detection was done, so all correlations are done on continuous traces. The disadvantage to this is that PV overlap could be affected by different brightness levels in different parts of the field of view

Note that the analyses are performed on DF/F traces, which normalizes for brightness. To ensure that our analysis approach didn't affect the results. We repeated the analysis for neurons using standard spike rate estimation (Fig S17). Although the data were somewhat noisier, the basic findings are similar to when we used the DF/F data.

3. I was confused with some aspects of figure 2. Cell 1 is described as exhibiting higher responses for context 1, but the responses seem to be clearly higher for context 2. Cell 2 is described as exhibiting higher responses for right turns, but this appears to be the case only for correct trials, though the effect is visible in some incorrect trials.

Our apologies, we had a typo in the figure caption. This has been corrected.

4. In Fig 3, are these cells pooled from several mice? Are there sufficient N to show similar patterns across mice?

The N for mice was too small for robust statistical comparisons (particular since some mice were excluded from each pair of days), so we instead used individual experiments (presented in Fig. 6).

5. In 5C, there is an additional distinction between post-decision outcome and post-trial outcome. Which of those is analogous to outcome in the other figures? Related, figure 5C possibly contradicts figure 3E. From 3E, I would expect outcome to be far more stable than context and motor, but in 5C, context is more stable than post-decision

outcome, and both are more stable than motor. Post-trial outcome is clearly the most stable, but I am unsure how to combine that with post-decision outcome.

We realized that this was confusing, so we have divided “outcome” into “post-decision outcome” and “post-trial outcome” throughout the manuscript.

6. It would be interesting to see if there was increased stability by a later stage of learning. All of the data presented is based on days with a handful of miss trials. Neighboring day analyses would also be helpful as a supplement. Also does 5 consecutive days seem like enough time to track stability. In the hippocampus studies track over longer time period (7-60 days). Please provide quantifiable measures of stability (PV Correlation; stability index; TC correlation, Representational Drift index)

Yes, these mice were already performing at plateau levels (we wanted to assess drift independently of learning). Although it would be interesting to analyze the stability during the learning phase, it often took animals quite some time to learn the task (>30 sessions), so this was not practical.

In terms of the longitudinal time course, a number of studies do examine hippocampus over shorter time periods and observe drift across single days (e.g., Hainmuller et al., *Nature*, 2018; Khatib, *Neuron*, 2023; Geva, *Neuron*, 2023). Compared to other cortical regions (particularly visual and motor cortex), retrosplenial cortex is quite unstable, which is why we are able to see changes over such a short time scale. Recent work comparing representational drift across cortical areas actually found the retrosplenial cortex to be one of the least stable regions (personal communication: Sofia Soares & Chris Harvey).

Minor:

1. In figure 3 the example cell shown for decoding motor choice seems to be very weakly modulated. Is there another possible example that has better performance?

We chose another example (Fig. 4A, cell 2) that had similar decoding performance to our example context and outcome neurons. We also include some additional examples in Fig S6.

2. Unclear why there are fewer cells in figure 3 than indicated in Figure 2B. If the numbers in figure 3 are correct, that implies that in Figure 2B there should be many cells classified as having no preference.

The cells included in the responses analysis (now Fig. 3B) were all cells that exhibit reliable responses during the task. The cells included in the decoding analysis (now Fig. 4) are only cells that exhibited significant decoding. This is a more stringent criteria, since some cells responded reliably but did not encode a particular task variable higher than chance.

3. Is there any topographical organization of cells with different response properties, as hinted at in Figure 5D? Do nearby neurons respond similarly, or is it a salt and pepper arrangement?

We did find some coarse topographical organization to the coding using single day responses, which reflected difference in input anatomy. This was reported in our previous paper (Franco & Goard, *Science Advances*, 2021; Figures 6-7), so we did not investigate it further in this manuscript.

REVIEWER COMMENTS

Reviewer #1 (Remarks to the Author):

The authors well addressed previous comments and the study is convincing and exciting. Here are a few minor additional points to potentially increase its value.

1. In Figure 1C, there is an interesting bias in mice toward left or right choices. Is it statistically significant and is there any observable dynamics during the learning process? In other words, if on day 1, a mouse is biased to the left, how will this bias evolve over days?

2. Figures 4C and 5C showed consistent pattern in average performance across all groups. However, using non-linear mixed-effect models and fitting an analytical function, could potentially result in more insights than comparing individual time points. For instance, in Fig. 5C's left panel, significant differences are observed only on the second day. Using a simpler model might reveal more noticeable distinctions.

3. In Figure 6D the authors applied a comparable approach. However, a more flexible model with 2-3 parameters and conducting computations with non-linear mixed-effect models may further enhance the accuracy. Given the use of Matlab, functions like `nlmefit/nlmefitsa` may be instrumental to implement this methodology effectively.

Typo: "Fig. 1. Context-choice association task for studying stability of cortical representations in mice. ...two-photon microscospe..."

Reviewer #4 (Remarks to the Author):

In this study, the authors compared the stability of task-related variables in RSC neural activity in mice performing a context-dependent virtual reality navigation task which they developed previously (Franco and Goard, *Science Advances* 2021). In their previous 2021 study, they found that RSC is necessary for decision making and encodes multiple task-related variables. In this new manuscript, they further investigated the coding in RSC activity by focusing on the coding stability over 5 days of consecutive behavior sessions. Specifically, they found that the post-trial outcome signal is most stable, the context and post-decision outcome signals are moderately stable, and the motor signal is least stable in RSC activity over the 5 expert sessions.

I am a new reviewer from this revision. I was asked to comment on the response of the authors to the previous Reviewer #2, so I will make these comments first. I have a few additional major concerns about this paper, which I will list at the end.

I think many of the concerns of Reviewer #2 were fully addressed by the authors. Some concerns (1-C, 4, 6) were only partially addressed.

1-C and 4: The Reviewer #2 had concerns about the consistency of their results across animals. This point is very important. However, the authors claim that the number of animals included in their data set (n=6 mice) is not sufficient for per-animal statistical analyses. Therefore, they re-analyzed their data using each imaging field separately. Although this approach is not ideal and leaves some concerns of the Reviewer #2, Fig.S13 where they plot the stability curves for individual mice is reassuring to some extent. Even if the statistical power is not enough, the authors should still include a plot like Fig6D grouped by animals in the Fig.S13. Also, the authors should use mixed-effects models for statistics when it is appropriate by accounting for the 'animal' effect for their statistics. The authors used mixed effects models for only some analyses.

Besides, the exact equation of the mixed effects model the authors used is unclear from their description. For example, 'linear mixed-effects model, fixed effects for task variable, random effects for mouse and experimental session'. Did their random effects include both 'mouse' and 'session' as separate terms, or did they include only their interaction (equivalent to 'session' only)? Did they include both random intercept and random slope? Additional details are necessary, and 'mouse' term should be included for all mixed effects models.

6: The Reviewer #2 requested stability analyses over days longer than 5 days and over learning phase. The authors did not directly address this concern. Authors claim it would not be practical to do the same analyses over a longer period. I believe it should be doable, but I also sympathize with authors on this point given the additional effort they need to collect new data. I agree that these data and analyses would be interesting but may not be necessary for this manuscript.

Additional major concerns:

The main finding in this manuscript is that the stability differs between different task-related signals in RSC. However, I am not fully convinced by the data currently provided in this manuscript.

1. For example, these stability analyses would be affected by the general activity level of neurons. For the comparisons between correct and incorrect trials (Fig2), the authors only mention in the discussion that 'we did not find differences in averaged activity levels between correct and

incorrect responses, except at the end of the trial' without showing the quantitative comparison in figure panels. Please include the activity comparisons for all trial periods for correct and incorrect trials. This is particularly important for this particular behavior task where mice do not need to physically move to proceed in the VR environment. In many incorrect trials, mice may not be simply engaged in the task well, leading to weaker neural activity. For trials with weaker activity, lower population stability is expected due to the noise effects. If the authors see differences in activity level between correct and incorrect trials, they should do the same analyses using only trials with matched mean activity levels for each trial period. The authors should also include comparisons of reaction time (time of joystick push) and joystick trajectory between correct and incorrect trials as the indirect measures of their task engagement.

2. Fig3-7 compares stability of 4 task-related signals. However, the stability seems to correlate very well with the strength of each task-related signal. Stability would look lower for weaker signals because the lower signal-to-noise ratio makes it more difficult to accurately estimate the true coding vector. A more inaccurate coding vector would negatively affect the stability measures. Therefore, it is not clear if the tau differences shown in the manuscript truly indicate the stability difference, or simply reflecting the signal strength. The authors should plot how well the strength of each signal correlates with the estimated decay constant tau. Furthermore, they should confirm that their main stability results are not affected by the signal strength. They can either do the same analyses by selecting only the experiments with matched signal strength, or artificially adding noise to neural activity to match the signal strength.

Minor comment:

1. Fig6C, 7C: The peak of an exponential function should be constrained to take the value of 1.

2. FigS3A: The legend says the peak sorting is cross-validated, but the plots do not look like cross-validated plots. They look very different from Fig2. Please confirm they are really cross-validated.

REVIEWER COMMENTS

The reviews from the previous round were generally positive, but there were a few remaining concerns, particularly around statistical approaches and controlling for initial coding strength. We have added a number of new analyses to address these issues. We thank the reviewers for their input, we believe these changes have significantly improved the strength of the paper.

Reviewer #1 (Remarks to the Author):

The authors well addressed previous comments and the study is convincing and exciting. Here are a few minor additional points to potentially increase its value.

1. In Figure 1C, there is an interesting bias in mice toward left or right choices. Is it statistically significant and is there any observable dynamics during the learning process? In other words, if on day 1, a mouse is biased to the left, how will this bias evolve over days?

We thank Reviewer #1 for pointing this out. We have now added the bias index (L-R/L+R) for each mouse through training and imaging sessions (Fig. S2C-D). Although one mouse shows a bias for right decisions during training, decision bias generally remains small from the end of training through the five days of imaging. The direction of bias shows stability across days (Fig. S2D).

2. Figures 4C and 5C showed consistent pattern in average performance across all groups. However, using non-linear mixed-effect models and fitting an analytical function, could potentially result in more insights than comparing individual time points. For instance, in Fig. 5C's left panel, significant differences are observed only on the second day. Using a simpler model might reveal more noticeable distinctions.

We completely agree, and tried using non-linear mixed-effects models. However, even after consulting with a statistician, we found that Matlab's *nlmefit* produced poor fits of our data. So, we developed an analysis to address the concern: we first fit the decoding performance (Figs. 4C, 5C, S12C) as well as the correlation coefficient (Fig. 3C, D) of each individual cell to a decay function (e.g., $\text{performance} = \text{performance}_{\text{day1}} * e^{(1-\text{day})/\text{tau}}$). We then extracted the fit parameters, rank transformed them (because they were not normally distributed), and compared them using linear mixed-effects models. We believe this analysis is a much improved strategy for comparing the stability of individual cells than our previous comparison between individual time points.

3. In Figure 6D the authors applied a comparable approach. However, a more flexible model with 2-3 parameters and conducting computations with non-linear mixed-effect models may further enhance the accuracy. Given the use of Matlab, functions like *nlmefit/nlmefitsa* may be instrumental to implement this methodology effectively.

As mentioned above, *nlmefit* did not produce good fits for our data set. So, similar to the approach described above, we first fit decay functions to each individual experiment (Fig. 6C-D), or to each mouse (Fig. S14C-D). Then, we compared their corresponding tau values using LME models.

Typo: “Fig. 1. Context-choice association task for studying stability of cortical representations in mice. ...two-photon microscope...”

Thank you, this typo has been corrected.

Reviewer #4 (Remarks to the Author):

In this study, the authors compared the stability of task-related variables in RSC neural activity in mice performing a context-dependent virtual reality navigation task which they developed previously (Franco and Goard, Science Advances 2021). In their previous 2021 study, they found that RSC is necessary for decision making and encodes multiple task-related variables. In this new manuscript, they further investigated the coding in RSC activity by focusing on the coding stability over 5 days of consecutive behavior sessions. Specifically, they found that the post-trial outcome signal is most stable, the context and post-decision outcome signals are moderately stable, and the motor signal is least stable in RSC activity over the 5 expert sessions.

I am a new reviewer from this revision. I was asked to comment on the response of the authors to the previous Reviewer #2, so I will make these comments first. I have a few additional major concerns about this paper, which I will list at the end.

I think many of the concerns of Reviewer #2 were fully addressed by the authors. Some concerns (1-C, 4, 6) were only partially addressed.

1-C and 4: The Reviewer #2 had concerns about the consistency of their results across animals. This point is very important. However, the authors claim that the number of animals included in their data set (n=6 mice) is not sufficient for per-animal statistical analyses. Therefore, they re-analyzed their data using each imaging field separately. Although this approach is not ideal and leaves some concerns of the Reviewer #2, Fig.S13 where they plot the stability curves for individual mice is reassuring to some extent. Even if the statistical power is not enough, the authors should still include a plot like Fig6D grouped by animals in the Fig.S13. Also, the authors should use mixed-effects models for statistics when it is appropriate by accounting for the ‘animal’ effect for their statistics. The authors used mixed effects models for only some analyses.

We thank reviewer #4 for joining the revision process and for providing new insights into our manuscript. As suggested, we have now added a new analysis, grouping data by mouse, and followed a similar approach as with the data for each experiment by separate. We find similar results, where context and post-trial outcome show higher stability than motor choice. However, this smaller data set did not allow us to see differences between post-decision outcome and the other task variables (as seen in our analysis with a larger data set for individual experiments).

In addition, we have now compared data groups using LME models whenever possible throughout the manuscript (i.e., when random effects terms such as experiment number or mouse number are present). For some of our analyses, we bootstrapped our data, and LME models cannot be used for statistical comparisons, so we used the 95% CIs instead.

Besides, the exact equation of the mixed effects model the authors used is unclear from their description. For example, 'linear mixed-effects model, fixed effects for task variable, random effects for mouse and experimental session'. Did their random effects include both 'mouse' and 'session' as separate terms, or did they include only their interaction (equivalent to 'session' only)? Did they include both random intercept and random slope? Additional details are necessary, and 'mouse' term should be included for all mixed effects models.

In the Methods section, under Statistical Information, we have expanded our explanation about how LME models were used. Briefly, we used Matlab's *fitlme* function. The equation generally used was 'response ~ group ID + (1|expID) + (1|mouseID)', and the fit method was 'REML'. The group ID is the fixed effect term of the particular data being compared, and the expID and/or mouseID are the random terms. The response is the particular measurement being compared (e.g. correlation coefficient, decoding performance).

6: The Reviewer #2 requested stability analyses over days longer than 5 days and over learning phase. The authors did not directly address this concern. Authors claim it would not be practical to do the same analyses over a longer period. I believe it should be doable, but I also sympathize with authors on this point given the additional effort they need to collect new data. I agree that these data and analyses would be interesting but may not be necessary for this manuscript.

We agree that it would be interesting to look at longer time scales for decoding stability. Unfortunately, the experimental cohort of mice has reached the experimental end point and the first author has moved to a new institution, so these experiments are beyond the scope of the current project.

Additional major concerns:

The main finding in this manuscript is that the stability differs between different task-related signals in RSC. However, I am not fully convinced by the data currently provided in this manuscript.

1. For example, these stability analyses would be affected by the general activity level of neurons. For the comparisons between correct and incorrect trials (Fig2), the authors only mention in the discussion that 'we did not find differences in averaged activity levels between correct and incorrect responses, except at the end of the trial' without showing the quantitative comparison in figure panels. Please include the activity comparisons for all trial periods for correct and incorrect trials. This is particularly important for this particular behavior task where mice do not need to physically move to proceed in the VR environment. In many incorrect trials, mice may not be simply engaged in the task well, leading to weaker neural activity. For trials with weaker activity, lower population stability is expected due to the noise effects. If the authors see differences in activity level between correct and incorrect trials, they should do the same analyses using only trials with matched mean activity levels for each trial period. The authors should also include comparisons of reaction time (time of joystick push) and joystick trajectory between correct and incorrect trials as the indirect measures of their task engagement.

We thank reviewer #4 for pointing this out. We have now included comparisons for activity levels between correct and incorrect trials in Fig. S5D, E. We observe that although a small fraction of neurons shows increased activity for either correct or incorrect decisions through trial duration (mostly in the reward period following the trial), most neurons maintain similar levels of activity.

We have also added the reaction time and joystick trajectories for correct and incorrect decisions in Fig. S2E-G. We do observe differences in reaction times. In particular, at the end of training and through imaging sessions, where mice generally make faster decisions on correct trials compared to incorrect trials.

2. Fig3-7 compares stability of 4 task-related signals. However, the stability seems to correlate very well with the strength of each task-related signal. Stability would look lower for weaker signals because the lower signal-to-noise ratio makes it more difficult to accurately estimate the true coding vector. A more inaccurate coding vector would negatively affect the stability measures. Therefore, it is not clear if the tau differences shown in the manuscript truly indicate the stability difference, or simply reflecting the signal strength. The authors should plot how well the strength of each signal correlates with the estimated decay constant tau. Furthermore, they should confirm that their main stability results are not affected by the signal strength. They can either do the same analyses by selecting only the experiments with matched signal strength, or artificially adding noise to neural activity to match the signal strength.

We agree, and this concerned us as well. We developed a new analysis that we believe addresses this concern directly. We first compared the performance for all four task variables using different sized samples of neurons, and used this to estimate the number of resampled neurons needed for each task variable to achieve 90% decoding performance in day 1 (Fig. S19C). Then, we recalculated population decoding stability using these subpopulations of neurons (Fig. S19D). We still observe higher stability for context and post-trial outcome, and lower stability for motor choice and post-decision outcome, indicating that the differential stability does not depend on the initial decoding accuracy.

Minor comment:

1. Fig6C, 7C: The peak of an exponential function should be constrained to take the value of 1.

We have corrected our analysis to make day 1 performance equal to 1 (as opposed to day = 0). For instance, for normalized performance (NP) in Fig. 7C:

$$NP = e^{\frac{1-\text{day}}{\tau}}$$

2. FigS3A: The legend says the peak sorting is cross-validated, but the plots do not look like cross-validated plots. They look very different from Fig2. Please confirm they are really cross-validated.

Thank you for catching this error. We realized that although trials are separated for each individual iteration during the resampling process, when we average across iterations (for the visualization in former Fig. S3A), it caused the day 1 responses to sample from overlapping sets of trials across iterations, thus disrupting the cross-validation.

This visualization was not necessary for the analysis, so we removed the resampled activity in previous Fig. S3A, and kept only the analyses for the correlation matrices as well as the correlation along the diagonal in current Fig. S4 (which is calculated on individual iterations prior to averaging). We double checked the other plots of neural activity. Since they do not involve resampling across trials, our cross-validation approach is valid for all other plots throughout the manuscript.

REVIEWERS' COMMENTS

Reviewer #1 (Remarks to the Author):

To deal with individual variability, the authors first fitted decay functions to each experiment or mouse and then compared the corresponding tau values using LME models. This is fully appropriate. They also performed other requested analyses and improved the description of the analyses performed, so this manuscript version is satisfactory and provides an accurate and convincing account of this interesting study.

Reviewer #3 (Remarks to the Author):

Reviewer #4 (Remarks to the Author):

The authors addressed my major concerns with this revision. I support the publication of this manuscript in Nature Communications, but I have some minor comments.

- The expanded explanation about the LME model in the method section is helpful, but it still does not explain whether the model used random intercept and/or random slope. Based on the equation in the rebuttal letter, the model used the experiment number and the mouse number as the random intercepts but not for random slopes. The authors should mention that they used them as the random intercepts in the method section.

- On page 7, Fig. S4D-E should be S5D-E.

REVIEWERS' COMMENTS

Reviewer #1 (Remarks to the Author):

To deal with individual variability, the authors first fitted decay functions to each experiment or mouse and then compared the corresponding tau values using LME models. This is fully appropriate. They also performed other requested analyses and improved the description of the analyses performed, so this manuscript version is satisfactory and provides an accurate and convincing account of this interesting study.

We thank reviewer #1 for addressing important issues about our manuscript, in particular with our analyses involving exponential decay functions and the statistical comparisons using LME models.

Reviewer #3 (Remarks to the Author):

We thank reviewer #3 for helping out during the review process of our manuscript.

Reviewer #4 (Remarks to the Author):

The authors addressed my major concerns with this revision. I support the publication of this manuscript in Nature Communications, but I have some minor comments.

We thank reviewer #4 for their valuable input, in particular for helping us provide a better explanation of our statistical comparisons using LME models, but more importantly for improving our exponential decay fits and important control analyses.

- The expanded explanation about the LME model in the method section is helpful, but it still does not explain whether the model used random intercept and/or random slope. Based on the equation in the rebuttal letter, the model used the experiment number and the mouse number as the random intercepts but not for random slopes. The authors should mention that they used them as the random intercepts in the method section.

The explanation about the LME model has been updated in the methods section. This is part of the updated description:

“[...] data groups are compared using linear mixed-effects (LME) models using task variables or trial type for the fixed effect for the intercept, and the experiment number

and the mouse number as nominal values for the random effects for the intercept [...]"

We hope this updated description is more accurate.

- On page 7, Fig. S4D-E should be S5D-E.

Thank you for catching this error. We have updated that reference to 'Supplementary Fig. 5d-e'.